

# Transition from hydrothermal vents to cold seeps records timing of carbon release in the Guaymas Basin, Gulf of California

Sonja Geilert[1], Christian Hensen[1], Mark Schmidt[1], Volker Liebetrau[1], Florian Scholz[1], Mechthild Doll[2], Longhui Deng[3], Mark A. Lever[3], Chih-Chieh Su[4], Stefan Schlömer[5], Sudipta Sarkar[6], Volker Thiel[7], Christian Berndt[1]

[1]GEOMAR Helmholtz Centre for Ocean Research Kiel, Wischhofstraße 1-3, 24148 Kiel, Germany
[2]Universität Bremen, Klagenfurter-Straße 4, 28359 Bremen, Germany
[3] Department of Environmental Systems Science, ETH Zurich, Universitätstrasse 16, 8092 Zurich, Switzerland
[4]Institute of Oceanography, National Taiwan University, No. 1, Sec. 4, Roosevelt Road, Taipei 106, Taiwan
[5]Federal Institute for Geosciences and Natural Resources, Stilleweg 2, 30655 Hannover, Germany
[6]Department of Earth and Climate Science, Indian Institute of Science Education and Research Pune, Dr. Homi Bhabha Road, Maharashtra-411008, India
[7]Geobiology, Geoscience Centre, Georg-August University Göttingen, Goldschmidtstr. 3, 37077 Göttingen, Germany

## Abstract

The Guaymas Basin in the Gulf of California is an ideal site to test the hypothesis that magmatic intrusions into organic-rich sediments can cause the release of large amounts of thermogenic methane and $CO_2$ that may lead to climate warming. In this study pore fluids close (~500 m) to a hydrothermal vent field and at cold seeps up to 20 km away from the northern rift axis were studied to determine the influence of magmatic intrusions on pore fluid composition and gas migration. Pore fluids close to the hydrothermal vent area show predominantly seawater composition, indicating a shallow circulation system transporting seawater to the hydrothermal catchment area rather than being influenced by hydrothermal fluids themselves. Only in the deeper part of the sediment core, composed of hydrothermal vent debris, Sr isotopes indicate a mixture with hydrothermal fluids of ~3%. Also cold seep pore fluids show mainly seawater composition. Most of the methane is of microbial origin and consumed by anaerobic oxidation in shallow sediments, whereas ethane has a clear thermogenic signature. Fluid and gas flow might have been active during sill emplacement in the Guaymas Basin, but ceased 28 to 7 thousand years ago, based on sediment thickness above extinct conduits. Our results indicate that carbon release depends on the longevity of sill-induced, hydrothermal systems which is a currently unconstrained factor.




## 1 Introduction


Climate change events in Earth's history have been partly related to the injection of large
amounts of greenhouse gases into the atmosphere (e.g. Svensen et al., 2004; Gutjahr et al.,
2017). One of the most prominent events was the Paleocene-Eocene Thermal Maximum
(PETM) during which the Earth's atmosphere warmed by about 8°C in less than 10,000 years
(Zachos et al., 2003). The PETM was possibly triggered by the emission of about 2000 Gt of
carbon (Dickens, 2003; Zachos et al., 2003). Processes discussed to release these large
amounts of carbon in a relatively short time are gas hydrate dissociation and igneous
intrusions into organic-rich sediments, triggering the release of carbon during contact
metamorphism (Kennett et al., 2000; Svensen et al., 2004). The Guaymas Basin in the Gulf of
California is considered one of the few key sites to study carbon release in a rift basin
exposed to high sedimentation rates.
The Gulf of California is located between the Mexican mainland and the Baja California
Peninsula, north of the East Pacific Rise (EPR; Fig. 1). The spreading regime at EPR continues
into the Gulf of California and changes from a mature, open ocean-type to an early-opening
continental rifting environment with spreading rates of about 6 cm yr$^{-1}$ (Curray & Moore,
1982). The Guaymas Basin, which is about 240 km long, 60 km wide, and reaching water
depths of up to 2000 m, is known as a region of vigorous hydrothermal activity (e.g. Curray
and Moore, 1982; Gieskes et al., 1982; Von Damm et al., 1985). Its spreading axis consists of
two graben systems (northern and southern troughs) offset by a transform fault (Fig. 1). In
contrast to open ocean spreading centres like the EPR, the rifting environment in the
Guaymas Basin shows a high sediment accumulation rate of up to 0.8-2.5 m kyr$^{-1}$ resulting in
organic-rich sedimentary deposits of several hundreds of meters in thickness (e.g. Calvert,
1966; DeMaster, 1981; Berndt et al., 2016). The high sedimentation rate is caused by high
biological productivity in the water column and influx of terrigenous matter from the
Mexican mainland  (Calvert, 1966).
Hydrothermal activity in the Guaymas Basin was first reported in the southern trough (e.g.
Lupton, 1979; Gieskes et al., 1982; Campbell and Gieskes, 1984; Von Damm et al., 1985).
Here, fluids emanate, partly from Black Smoker type vents at temperatures of up to 315°C





(Von Damm et al., 1985). Sills and dikes intruding into the sediment cover significantly affect
temperature distribution, and hence environmental conditions (Biddle et al., 2012; Einsele et
al., 1980; Kastner, 1982; Kastner and Siever, 1983; Simoneit et al., 1992; Lizarralde et al.,
2010; Teske et al., 2014). The magmatic intrusions accelerate early-diagenetic processes and
strongly influence the chemistry of the interstitial waters (e.g. Gieskes et al., 1982; Brumsack
and Gieskes, 1983; Kastner and Siever, 1983; Von Damm et al., 1985). Lizarralde et al. (2010)
reported that sills intruded into the sediment cover and that cold seeps at the seafloor are
visible up to 50 km away from the rift axis. They proposed a recently active magmatic
process that released much higher amounts of carbon into the water column than previously
thought. It was assumed that magmatic intrusions trigger the alteration of organic-rich
sediments and release thermogenic methane and $CO_2$. Varying methane concentrations and
temperature anomalies in the water column were interpreted as active thermogenic
methane production generated by contact metamorphism (Lizarralde et al., 2010). This
process might cause a maximum carbon flux of 240 kt C $yr^{-1}$ and might induce profound
climatic changes.
During the SO241 expedition in June/ July 2015 a new hydrothermal vent field was
discovered at the flank of the northern trough (Fig. 1; Berndt et al., 2016). The discovered
mound rises up to 100 m above the seafloor and predominantly Black Smoker type vents
suggest similar endmember temperatures and geochemical composition as found at the
southern trough (Berndt et al., 2016; von Damm et al. 1985). Berndt et al. (2016) discovered
an active hydrothermal vent system comprised of black smoker-type chimneys that release
methane-rich fluids with a helium isotope signature indicative of mid-ocean ridge basalt. The
vigorous release of large amounts of methane and $CO_2$ several hundred of meters into the
water column combined with magmatic intrusions into underlying sediments led Berndt et
al. (2016) to support the hypothesis that this process might have triggered the PETM during
opening of the North Atlantic as proposed by Svensen et al. (2004).
During RV SONNE cruise SO241, both, the recently discovered hydrothermal vent in the
northern trough (Berndt et al., 2016) and some of the off-axis seeps (Lizarralde et al. 2010)
which are located above potential sill intrusions were investigated by sediment, carbonate,
and water column sampling. Here, we present fluid and gas geochemical data from both





systems as well as carbonate data and discuss these data in the context of seismic data in
order to constrain subsurface processes and fluid origin.

**2 Materials and methods**
2.1 Sampling devices and strategy

During the RV SONNE expedition SO241 seven sites across the central graben of the
Guaymas basin were investigated (Fig. 1). Site-specific sampling and data recording was
performed using a (1) video-guided multicorer (MUC), (2) gravity corer (GC), (3) temperature
loggers attached to the GC or sediment probe, (5) CTD / Rosette water sampler, and (6)
video-guided hydraulic grab (VgHG). Sites were selected according to published data on the
location of seeps (Lizarralde et al., 2010) and seismic data acquired during the cruise (see
below).

2.1.1 Seismic data recording

Seismic data were collected using a Geometrics GeoEel Streamer of 150 to 183.5 m length
and 96 and 112 channels, respectively. Two GI guns in harmonic mode (105/105 cubic inch)
served as the seismic source. Processing included navigation processing (1.5625 m crooked
line binning), 20, 45, 250, 400 Hz frequency filtering, and poststack Stolt migration with
water velocity yielding approximately 2 m horizontal and 5 m vertical resolution close to the
seafloor.

2.1.2 Sediment and pore fluid sampling

At seepage and vent sites, the video-guided multicorer was used to discover recent fluid
release, which is indicated by typical chemosynthetic biological communities at the seafloor
(bacterial mats, bivalves, etc.). However, small-scale, patchy distributions of active seepage
spots and visibility of authigenic concretions made it difficult to select the "best possible"
sampling locations for getting fine-grained sediment samples. Hence, comparing results from
different seeps might be biased in this regard. GC deployments were typically performed at





pre-inspected MUC sites or at the center of suspected seeps (based on bathymetry and
seismic data).
In total, we present pore fluid data collected at three seepage sites, North (GC01, MUC11),
Central (GC03, GC13, GC15, MUC04), and Ring Seeps (MUC05), one Reference Site (no active
seep site, see definition above; GC04, MUC02), and one active hydrothermal site, Smoker
(GC09, GC10, MUC15, MUC16). A Reference Site, that did not show active seepage or faults
indicated by seismic data, was chosen to obtain geochemical background values. In addition,
the slope towards the Mexican mainland was sampled as well (GC07) (Fig. 1, Table 1). After
core retrieval, gravity cores were cut and split on deck and immediately sampled. Samples
were transferred into a cooling lab at 4°C and processed within 1 or 2 hours. Pore fluids were
obtained by pressure filtration. Sediment samples for hydrocarbon gases were taken on deck
with syringes and transferred to vials containing concentrated NaCl solution (after Sommer
et al., 2009). After multicorer retrieval, bottom water was sampled and immediately filtered
for further analyses. The sediment was transferred into a cooling lab and sampling was
executed in an argon-flushed glove bag. Pore fluids were retrieved by centrifugation and
subsequent filtration using 0.2 µm cellulose acetate membrane filters.

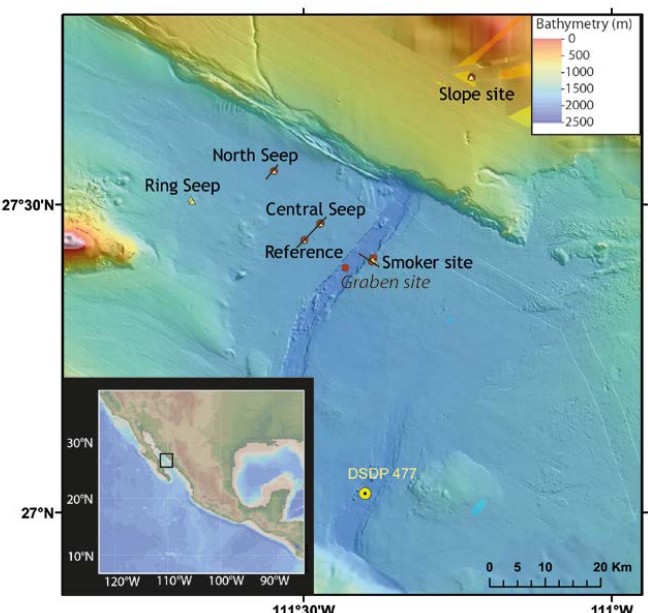






Figure 1: Sample locations in the Guaymas Basin, Gulf of California, during RV SONNE
expedition SO241. Black lines refer to seismic profiles, displayed in Fig. 2. Graben Site refers
to water column sampling only.

2.1.3 Subseafloor temperature measurements

Temperature gradients and thermal conductivity were measured at North Seep, Central
Seep, Reference Site, and Smoker Site as well as along a transect across the newly
discovered hydrothermal vent field and the rift valley. Miniaturized temperature loggers
(MTL) were attached to gravity cores or to a 5 m long sediment lance at a sampling rate of 1
s. The absolute accuracy of these temperature measurements is about 0.1 K and the
temperature resolution is 0.001 K (Pfender and Villinger, 2002).
Thermal conductivity was measured on recovered core material in close vicinity to the MTLs
using the KD2 Pro Needle Probe instrument. For temperature measurements obtained by a
lance, a constant thermal conductivity of 0.7 W/m K was assumed. Data processing was
done according to Hartmann and Villinger (2002).

2.1.4 Water column sampling

Water samples were taken by using a video-guided Niskin Water sampler Rosette System
(Schmidt et al., 2015) in order to study water column chemistry (i.e. dissolved $CH_4$) and
oceanographic parameters (i.e. temperature, salinity, turbidity). Eight water sampling
locations were chosen in the vicinity of MUC and GC stations and are termed North
(VCTD03), Central (VCTD02), Ring (VCTD01), Graben (CTD01; no video-guided sampling),
Smoker (VCTD06 and 10), and Slope (VCTD07). Additionally, hydrocarbon data published in
Berndt et al. (2016) from the Smoker Site (VCTD09) are shown. The (V)CTDs were either used
in a towed mode (VCTD03, 06, 09, 10) or in station (CTD01; VCTD01, 02, 07) keeping
hydrocast mode. The water depth was controlled based on pressure readings, altitude
sensors (<50 m distance to bottom), and online video observation (1 - 2 m above the
seafloor).

2.1.5 Authigenic carbonate sampling






At Central Seep a block (approx. 1 x 0.5 x 0.3 m) mainly consisting of solidified carbonate
matrix covered by a whitish carbonate rim and characterized by coarse open pore space in
mm to cm scale (see supplementary Fig. 1S) was recovered in 1843 m water depth from the
surface of a typical cold seep environment (close to high abundance of tube worms) by the
deployment of a video-guided hydraulic grab (VgHG, GEOMAR).

185          2.2 Sample treatment and analytical procedures

Pore fluids were analyzed onboard by photometry (hydrogen sulfide and $NH_4$) and titration
(total alkalinity = TA). Subsamples were analyzed in shore-based laboratories for major
anions and cations using ion chromatography (IC, METROHM 761 Compact, conductivity
mode) and inductively coupled plasma optical emission spectrometry (ICP-OES, VARIAN 720-
ES), respectively. Detailed descriptions can be found elsewhere (e.g. Scholz et al., 2013). All
chemical analyses were tested for accuracy and reproducibility using the IAPSO salinity
standard (Gieskes et al., 1991).
Strontium isotope ratios were analyzed by Thermal Ionization Mass Spectrometry (TIMS,
Triton, ThermoFisher Scientific). The samples were chemically separated via cation exchange
chromatography using the SrSpec resin (Eichrom). The isotope ratios were normalized to
NIST SRM 987 value of 0.710248 (Howarth and McArthur, 2004) which reached a precision
of ± 0.000015 (2 sd, n = 12).
Water samples taken from Niskin bottles were transferred into 100 ml glass vials with helium
headspace of 5 ml and poisoned with 50 µl of saturated mercury chloride solution.
Hydrocarbon composition of headspace gases was determined using a CE 8000 TOP gas
chromatograph equipped with a 30 m capillary column (Restek Q-PLOT, 0.32 mm) and a
flame ionization detector (FID). Replicate measurements yielded a precision of <3% (2 sd).
Stable carbon isotopes of methane were measured using a continuous flow isotope ratio
mass spectrometer (cf-IRMS). A Thermo TRACE gas chromatograph was used to separate the
light hydrocarbon gases by injecting up to 1 ml headspace gas on a ShinCarbon ST100/120
packed gas chromatography column. The separated gases were combusted and



corresponding $\delta^{13}C$ values were determined using a Thermo MAT53 mass spectrometer. The
reproducibility of $\delta^{13}C$ measurements was ±0.3‰ VPDB (2 sd).
Stable hydrogen isotope compositions of methane were analyzed by separating methane
from other gases by online gas chromatography (Thermo Trace GC; isotherm at 30°C; 30 m
RT-Q-Bond column, 0.25 mm ID, film thickness 8 μm). Prior to stable isotope analysis using a
coupled MAT 253 mass spectrometer (Thermo) methane-H was reduced to dihydrogen at
1420°C. Data are reported in per mil relative to Standard Mean Ocean Water (SMOW). The
precision of $\delta D$-$CH_4$ measurements was ±3‰ (2 sd).

$^{210}Pb$ (46.52 keV) and $^{214}Pb$ (351.99 keV) were simultaneously measured by two HPGe
gamma spectrometry systems (ORTEC GMX-120265 and GWL-100230), each interfaced to a
digital gamma-ray spectrometer (DSPecPlus™). Efficiency calibration of the gamma detectors
were calibrated using IAEA reference materials, coupled with an in-house secondary
standard for various masses (Huh et al., 2006; Lee et al., 2004). $^{214}Pb$ was used as an index of
$^{226}Ra$ (supported $^{210}Pb$) whose activity concentration was subtracted from the total $^{210}Pb$ to
obtain excess $^{210}Pb$ ($^{210}Pb_{ex}$). The activities of radionuclides were decay-corrected to the date
of sample collection. All radionuclide data are calculated on salt-free dry weight basis.

A representative sample of the authigenic carbonate (cm-scale) was broken from the upper
surface of the block, gently cleaned from loosely bound sediment and organic remains and
dried at 20°C for 12 hrs. Two different subsamples were prepared by drilling material with a
handheld mm-sized mini-drill from the outer rim (whitish coating, lab code: 470-15) and the
related inner core (dark matrix, lab code: 472-15).
Prior to aliquot procedures both subsamples were finely ground in an agate mortar
providing homogeneous aliquots of suitable grain size for the combined approach of mineral
identification by X-ray diffractometry (XRD) (Philips X-ray diffractometer PW 1710 in
monochromatic CuKα mode between 2 and 70 2θ (incident angle), for details see
supplement), $\delta^{18}O$ and $\delta^{13}C$ analyses by stable isotope ratio mass spectrometry (SIRMS) and
U-Th geochronology by multi collector-inductively coupled plasma-mass spectrometry (MC-
ICP-MS) on a parallel leachate / sequential dissolution approach for single and isochron ages
(method see supplement) as well as $^{87}Sr$/$^{86}Sr$ isotope signatures for aliquots of the individual
U-Th solutions by thermal ionization mass spectrometry (TIMS, for method details please



refer to pore water Sr isotope analyses). Lipids extracts for biomarker determination were
analyzed as well (see below).

From each homogenized carbonate powder sample (see above), an aliquot of 10 mg was
separated for carbon $\delta^{13}$C and oxygen $\delta^{18}$O stable isotope analysis. A fraction from this
(approximately 1 mg) was dissolved by water-free phosphoric acid at 73°C in a "Carbo-Kiel"
(Thermo Fischer Scientific Inc.) online carbonate preparation line and measured for carbon
and oxygen stable isotope ratios with a MAT 253 mass spectrometer (Thermo-Fischer Inc.).
The $\delta^{13}$C and $\delta^{18}$O values are calculated as deviations from laboratory standard referred to
the PDB scale and reported in ‰ relative to V-PDB. The external reproducibility was checked
by replicate analyses of laboratory standards as being better than ±0.04‰ for $\delta^{13}$C and
±0.1‰ for $\delta^{18}$O (1SD, n=7) for this sample set. However, the single measurement
uncertainties were significantly better and the resulting 2SD (n=3) for both main samples are
given in the supplement table S5.

Biomarkers were determined by grounding 4g of the sample and were then sequentially
extracted with dichloromethane (DCM)/methanol (3/1, v/v), DCM, and n-hexane
(ultrasonication, 20 min). The combined extracts were dried, derivatized using a
BSTFA/trimethylchlorosilane mixture (95/5, v/v; 1h; 40°C) and analysed by coupled gas
chromatography-mass spectrometry (GC-MS). GC-MS analyses were carried out with a
Thermo Fisher Trace 1310  GC coupled to a Quantum XLS Ultra MS. The GC was equipped
with a Phenomenex Zebron ZB 5MS capillary column (30 m, 0.1 μm film thickness, inner
diameter 0.25 mm). Fractions were injected splitless at 270°C. The carrier gas was He (1.5
mL/min). The GC oven temperature was ramped from 80°C (1 min) to 310°C at 5°C min-1
and held for 20 min. Electron ionization mass spectra were recorded at 70 eV.

**3 Results**
3.1 Subsurface structure and evidence for sill-related fluid mobilization

Seismic profiles show a wide range of sediment deformation (Fig. 2). Seismic amplitude
blanking along vertical zones below the seafloor indicates apparent fluid flow at North,
Central, and Ring Seep (Fig. 2). Underneath these locations, sediments are deformed.





Blankening of the seismic signal is attributed to sediment mobilization due to the
hydrothermal activity in response to sill intrusion. In contrast, at the Reference Site
sediments show a more or less continuous succession without vertical disturbance. At North
Seep, a shallow high-amplitude reversed polarity reflector occurs at 50-60 mbsf. Sill depths
are inferred from the seismic profiles at ~500 to 600 m for North Seep and with ~350 to 400
mbsf at the other sites, assuming seismic interval velocities of 1600 to 2000 m s$^{-1}$. Seismic
images suggest that massive disturbance of sediments and vertical pipe structures are
related to channeled fluid and/or gas advection caused by sill intrusions (Fig. 2). Faults are
indicated which may serve as fluid pathways above potential sill intrusions. Closer inspection
of the seismic reflectors at the Central Seep (Fig 2c) shows onlap onto a doming structure.
On the NW flank of the dome the deepest onlap occurs at 60 ms or 48 m below the sea floor
(assuming 1600 m s$^{-1}$ sediment interval velocity) whereas on the SE flank the shallowest
onlap occurs at 15 ms or 12 m below the sea floor.

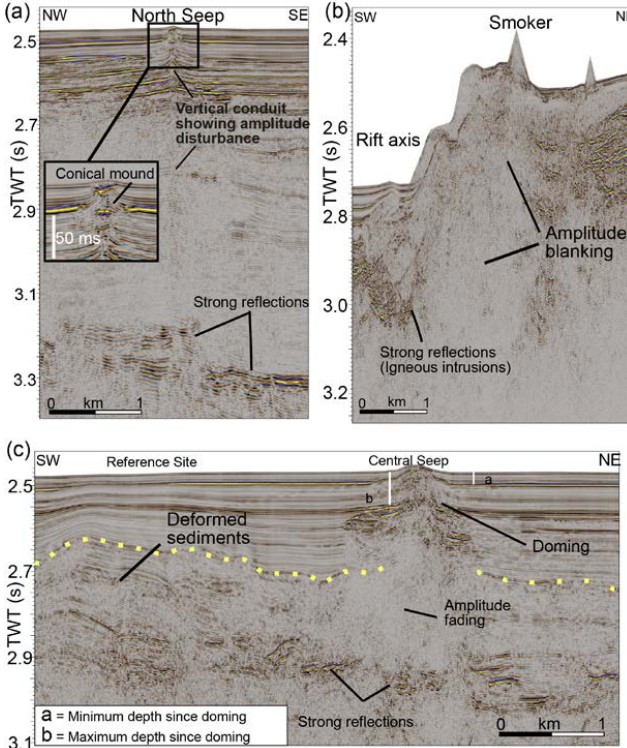


Fig.2: Seismic profiles of North Seep (a), Smoker Site (b) as well as of Central Seep and
Reference Site (c). Seismic section showing doming above the Central Seep. There are





different phases of onlap starting about 60 mbsf (maximum deposition) until about 15 mbsf
(minimum deposition) or 48 and 12 mbsf respectively assuming a sediment interval velocity
of 1600 m s$^{-1}$.

292        3.2 Temperature measurements


Heat flow and temperature gradients were measured at North and Central Seep, Reference
Site, and Smoker Site (attached to GCs) as well as in transects along the hydrothermal ridge
and rift axis (attached to a temperature lance; Fig. 3, Table 1). Highest heat flow values
occurred close to the Smoker Site and range between 599 and 10835 mW m$^{-2}$. Temperature
gradients were also highest at the Smoker Site (~15 K m$^{-1}$). In contrast, heat flow values and
temperature gradients in the rift valley close to the rift axis ranged between 262 and 338
mW m$^{-2}$ and 0.4 to 0.5 K m$^{-1}$, respectively. Generally heat flow values decreased with
increasing distance to the rift axis with 140 mW m$^{-2}$ at the Reference Site, 113 mW m$^{-2}$ at
Central Seep, and 28 mW m$^{-2}$ at North Seep. Temperature gradients are 0.22 K m$^{-1}$ at the
Reference Site, 0.16 K m$^{-1}$ at Central Site and 0.14 K m$^{-1}$ at North Site.

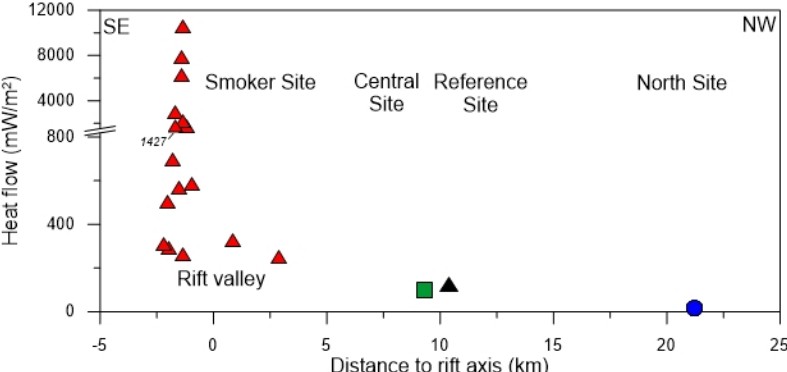


Figure 3: Heat flow in the Guaymas Basin in relative distance to the rift axis.

308        3.3 Sediment characteristics and sedimentation rates


The sediments are mainly composed of organic-rich diatomaceous clay, consistent with
earlier analyses (e.g. Kastner, 1982). At North Seep, the sediments are composed of
homogeneous diatomaceous clay. Rare shell fragments and carbonate concretions are



present. Gas hydrates were discovered at 2.5 meters below seafloor (mbsf). Authigenic
carbonates were present at the seafloor. At Ring Seep, SW of North Seep, sediments are
predominantly composed of diatomaceous clay. At Central Seep, located between North
Seep and Smoker Site, sediments are composed of homogeneous diatomaceous clay
intercalated with shell fragments and banding of whitish layers in the lower meter of the GC.
At the seafloor, authigenic carbonates were present as well. At Smoker Site, ca. 500 m SE of
the hydrothermal vent field, surface sediments are likewise composed of diatomaceous clay
with light and dark greyish banding. Traces of bioturbation are visible in the upper 4 m.
Below about 4 m depth, a sharp contact defines the transition to hydrothermal deposits,
which are composed of mm-to-cm sized black to grey Fe-rich sulfides (for a detailed
description see Berndt et al. (2016)). Within the hydrothermal deposits brownish to grey clay
lenses appear. At the Slope Site, sediments are laminated in the mm- to cm-range. The
sediment is dominated by diatomaceous clay and only a few ash lenses exist.
The sedimentation rates ranged between 0.4 m kyr$^{-1}$ at Smoker Site and 3.5 m kyr$^{-1}$ at North
seep based on radionuclides measurements (Table 1). Sedimentation rates at all other sites
are about 2 m kyr$^{-1}$.

3.4 Pore water geochemistry

All pore water data and isotope measurements of $^{87}$Sr/$^{86}$Sr are listed in supplementary table
S2. Pore water profiles of alkalinity, $H_2S$, $SO_4^{2-}$, $CH_4$, $NH_4$, $Cl^-$, Mg, and Li are shown in Fig. 4a
(GCs) and 4b (MUCs).




Figure 4: Pore water profiles of GCs (a) and MUCs (b). Endmember composition of hydrothermal solutions from Von Damm et al. (1985) and hydrothermal plume geochemical composition from Berndt et al. (2016) are shown as well in (a).



Pore water constituents plotted in Figure 4 were selected to characterize variations in
organic matter diagenesis, anaerobic oxidation of methane (AOM), as well as potential
water-rock interactions related to subsurface hydrothermal activity. In general, methane
concentrations are elevated at the seep locations and at the slope, thus enhancing AOM.
Alkalinity and $H_2S$ increase with depth for North Seep, Central Seep, and Slope Site, while
$SO_4^{2-}$ is decreasing. AOM depths can only be inferred for North Seep with ~160cm and Slope
Site with ~300cm. $NH_4$ is only slightly increasing with depth; higher $NH_4$-levels are only found
at the Slope Site (Fig. 4). Concentrations of $Cl^-$, Mg, and Li do not show significant variations
from seawater.
Sr concentrations and isotopes are plotted in Fig. 5. Sr concentrations show predominantly
modern seawater values, except at North Seep where they strongly decrease. The $^{87}Sr/^{86}Sr$
isotope ratios also show predominantly seawater values (0.709176; Howarth and McArthur,
2004). North and Ring Seeps show slight decreases in $^{87}Sr/^{86}Sr$, whereas values at the
Smoker Site decrease strongly below the transition between hemipelagic sediments and
hydrothermal deposits (Fig. 5). The ratios show a similar depletion as those from the
hydrothermal plume (Berndt et al., 2016).

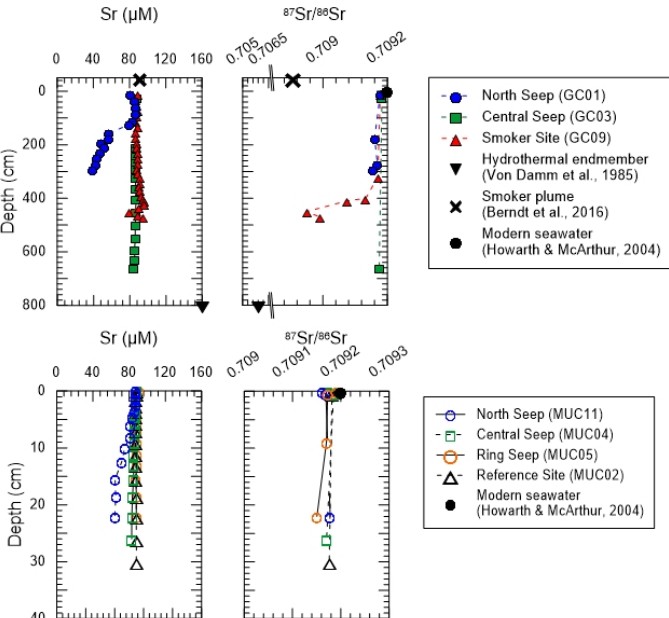




**Fig. 5.** Sr concentrations and $^{87}$Sr/$^{86}$Sr ratios for GCs (upper panels) and MUCs (lower panels).
For comparison, data from the hydrothermal smoker plume (Berndt et al., 2016), the
hydrothermal endmember (Von Damm et al., 1985), and modern seawater (Howarth and
McArthur, 2004) are shown in the upper panel. Note the different scale for MUC $^{87}$Sr/$^{86}$Sr
ratios.

3.5 Pore water hydrocarbon gases, carbon and hydrogen isotope data

Concentrations of dissolved hydrocarbons and $\delta^{13}C_{CH4}$, $\delta^{13}C_{C2H6,}$ and $\delta D_{CH4}$ data are reported
in supplementary table S3. Overall, our data show a large variability in $CH_4/(C_2H_6+C_3H_8)$ with
ratios between 100 and 10,000 and $\delta^{13}C_{CH4}$ between -25 and -90 ‰. The $\delta^{13}C_{C2H6}$ values
range between -26.1 and -38.3 ‰ for North Seep and -29.6 and -37.7 ‰ for Central Seep.
The $\delta D_{CH4}$ values at both seeps range between -97 and -196 ‰, for Slope Site between -192
and -196 ‰, and for the Smoker hydrothermal plume between -98 and -113 ‰.

3.6 Water column data

Water column characteristics like temperature, salinity, turbidity as well as methane
concentrations are shown in figure 6 and supplementary table S4. Surface waters in the
Guaymas Basin show warm temperatures up to 29.5°C (salinity: 34.5‰) close to the Mexican
mainland (Slope, VCTD07) and up to 24.6°C (34.6‰) in the central basin (Central, VCTD02).
With depth, temperatures decrease continuously to 2.8 to 3.0°C (salinity: 34.6‰) close to
the sea floor (1600 - 1800m). Turbidity values are high in the deep water layer (~1400-
1800m) and indicate a well-mixed deep basin, also shown by relatively homogeneous
temperature and salinity data. Only the water column directly above the hydrothermal
smoker field (VCTD09) shows strongly elevated temperature (28.4°C) and salinity (35.1‰)
(Berndt et al., 2016). Methane concentrations are highest close to the smoker vent field (up
to 400 µM, (VCTD09; Berndt et al., 2016)), but still vary in the deep water column of the
basin between 2 and 28.1 nM (Central (VCTD02) and Ring (VCTD01), respectively).





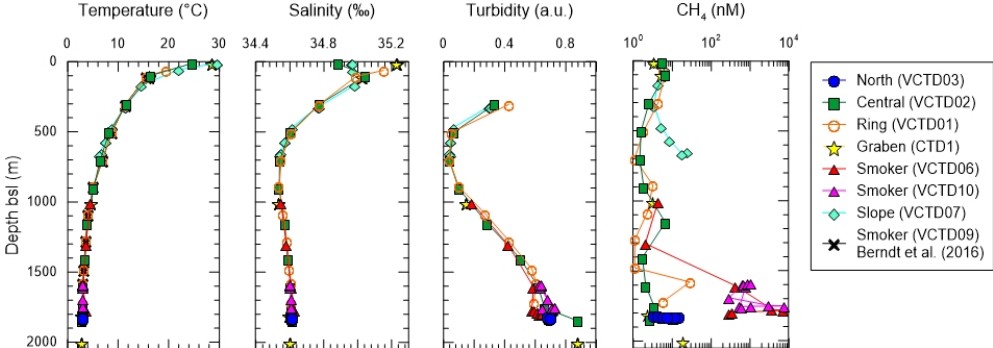

Fig.6: Water column temperature, salinity, turbidity, and methane concentrations. Note that the upper ~300m bsl in the turbidity data are not shown for scale matters. VCDT10 temperature data are from Berndt et al. (2016).

### 3.7 Authigenic carbonate data

The authigenic carbonate sample (Fig. S1) consists of 88 to 90 % aragonite and 6 to 12 % calcite (supplementary Table S5). The bulk outer rim carbonate has an average carbon isotope signature ($\delta^{13}C_{V\text{-}PDB}$) of -46.6±0.2‰ and an oxygen isotope signature ($\delta^{18}O_{V\text{-}PDB}$) of 3.7±0.3 ‰. Inner core carbonate isotope signatures yield similar values with $\delta^{13}C_{V\text{-}PDB}$ of -44.7±0.2 ‰ and $\delta^{18}O_{V\text{-}PDB}$ of 3.6 ±0.1 ‰ (Table S5). The average outer rim $^{87}Sr/^{86}Sr$ ratio is 0.709184 and the inner core ratio is 0.709176. External reproducibility of NIST-SRM987 is 0.000015 (2 SEM). The U-Th carbonate dating approach on these authigenic carbonates implies formation ages younger than 240 yrs BP.

Lipids extracts obtained from seep carbonate 56-VgHG-4 (Central Site) revealed a strong signal of specific prokaryote-derived biomarkers (Fig. S1). These compounds encompassed archaeal isoprenoid lipids, namely crocetane, 2,6,10,15,19-pentamethylicosane(-icosenes (PMI, PMIΔ) archaeol, and *sn*2-hydroxyarchaeol (see Fig. S1 for structures). In addition, the sample contained a suite of non-isoprenoid 1,2-dialkylglycerolethers (DAGE) of bacterial origin. Typical compounds of planktonic origin, such as sterols, were also present, but low in abundance.





## 4 Discussion

4.2 Origin of seeping fluids

4.2.1 Black Smoker Site

The water column above the newly discovered vent exhibits elevated $CH_4$ concentrations (up to 400 µM) and $pCO_2$ data (>6000 µatm), and the range of measured stable isotope signature of methane ($\delta^{13}C_{CH4}$ between -39‰ and -14.9‰) and a Helium ($^3He$) isotope anomaly clearly indicates gas exhalations from thermogenic organic matter degradation with contributions from a mantle source (Berndt et al., 2016). These northern trough hydrothermal fluids are comparable in their gas geochemistry to the southern trough (Lupton, 1979; Von Damm et al., 1985) as was demonstrated by endmember calculations in Berndt et al., 2016. However, the highest heat flow values up to 10835 mW/m$^2$ are found close to the Smoker Site and are much higher than observed in earlier studies in which maximal 2000 mW/m² were measured in the center of the trough (Fisher and Becker, 1991). The high heat flow at Smoker Site even exceeds the hydrothermally more active southern trough where heat flow values of 2000 to 9000 mW/m² were measured (Fisher and Becker, 1991; Lonsdale and Becker, 1985). This might indicate that hydrothermal activity at the northern trough is younger and a more recent process compared to the southern trough.

Despite the proximity of the gravity cores (GC09, GC10) and multicorer-cores (MUC15, MUC16) to the hydrothermal vent field (~500m distance; temperatures measured immediately after retrieval are up to 60°C) pore fluid geochemical signatures within nearby sediments are not much different from those in seawater (Fig. 4). Specifically Mg, Li, Cl, and $^{87}Sr/^{86}Sr$ which are considered as good indicators for hydrothermal alterations and/or deep-seated diagenetic processes do not show any prominent excursions from seawater values. Hydrothermal fluids are typically depleted in Mg and highly enriched in fluid-mobile elements like Li caused by high-temperature reactions with mafic rocks (here sills) and/or sediments through which they percolate (e.g. Einsele et al., 1980; Gieskes et al., 1982; Kastner, 1982; Von Damm et al., 1985; Lizarralde et al., 2010; Teske et al., 2016). Such compositions are reported from DSDP site 477 (Gieskes et al., 1982) and fluids obtained by Alvin dives (Von Damm et al., 1985). Although strongly diluted, CTD samples from the Black Smoker plume in the Northern trough show this trend (Berndt et al., 2016). Our data





therefore suggest that the sediments surrounding the Black Smoker area are not percolated
by hydrothermal fluids. We hypothesize that hydrothermal venting causes a shallow
convection cell (e.g. Henry et al., 1996) drawing seawater through the sediments towards
the smoker, while the sediments become heated by lateral heat conduction.
Geochemical indicators for a diagenetic or catagenetic breakdown of organic matter like $NH_4$
are only poorly enriched in sediments surrounding the black smoker vents. Expected end-
member values should be similar to those reported from the southern trough (20mM; Von
Damm et al. (1985)), but they remain well below (≤ 0.3mM). For comparison, intense organic
matter breakdown occurs in areas with high sediment accumulation rates like the
continental slope (Simoneit et al., 1986). Here, maximum $NH_4$-levels of 1-10 mM
(accompanied by high levels of alkalinity and AOM; Fig. 3) are reached in the pore water
already at subsurface depths of only a few meters, confirming that a fluid mobilized from
greater subsurface depth must be enriched in $NH_4$ and other products of organic matter
degradation. Overall, this confirms that early-diagenetic processes are not intense around
the Smoker Mound and further indicates a shallow convection mixing seawater into the
sediments in ≤4m depth.
Interestingly, there is a slight positive Li excursion at about 4 m depth in core GC09. This
might be related to the mineralogy of this sediment section where the main composition
changes from diatomaceous clay to hydrothermal deposit (Fe-rich sulfides; see also Sect.
3.3). We suspect that the positive Li anomaly is caused by weak admixing of hydrothermal
solutions, as none of the other elements shows drastic concentration changes indicative of
early-diagenetic reactions (Gieskes et al., 1982; Chan et al., 1994; Środoń, 1999; Chan and
Kastner, 2000; Aloisi et al., 2004; Hensen et al., 2007; Wallmann et al., 2008; Scholz et al.,
2009; 2010; 2013). Along with increasing Li concentrations, $^{87}Sr/^{86}Sr$ isotope ratios decrease
to a value of 0.70908 (Fig.5) and thus tend towards the $^{87}Sr/^{86}Sr$ ratio of the local
hydrothermal endmember ($^{87}Sr/^{86}Sr$ = 0.7059; Von Damm, 1990). Hydrothermal endmember
Li concentrations in the Guaymas Basin range between 630 and 1076 µM (Von Damm et al.,
1985) and are thus 20 to 30 times higher than the Li concentrations measured at the lower
end of the core at the Smoker Site (~34 µM; Fig. 4, Table S2) indicating a mixing between
seawater and hydrothermal fluids with a hydrothermal component of about ~3% (Fig. 7).




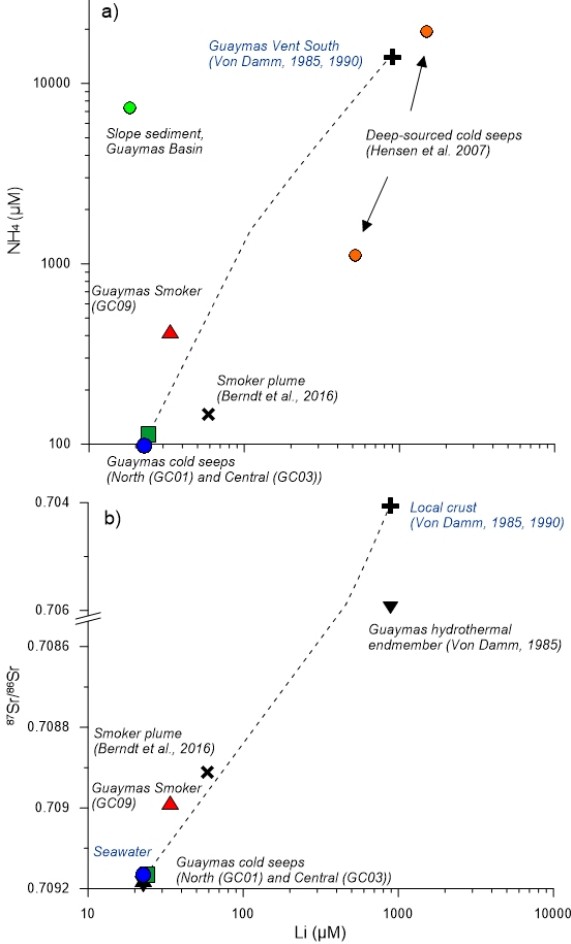


Fig. 7: NH$_4$ (µM) (a) and $^{87}$Sr/$^{86}$Sr ratios (b) versus Li concentrations (µM) of Guaymas Basin
cold seeps (North, Central) and the hydrothermal Smoker vent field. Guaymas deep Smoker
fluids (GC09) mix with hydrothermal fluids with a share of ~3%. For comparison, Guaymas
hydrothermal endmember fluid composition (Von Damm, 1985, 1990), Smoker plume fluid
composition (Berndt et al., 2016), slope sediments (in (a) and deep-sourced cold seeps from
the Gulf of Cadiz (in (a); Hensen et al., 2007)) are shown.

The hydrothermal activity in the northern trough of the Guaymas Basin can be summarized
to occur only in a relatively confined area affecting the surrounding sediments in a minor
way by lateral heat transfer. The diatomaceous clay might act as a seal to upwards migrating





fluids, which are channeled to the catchment area of the rising hydrothermal fluids of the
Black Smoker vent field (Fig. 4 in Berndt et al., 2016). The geochemical composition of the
upwards migrating hydrothermal fluids is likely influenced by high temperature chemical
alteration reactions between the sediment and the intruded sills (Fig. 2b). However,
shallower pore fluids of surface sediments at the smoker site (i.e. 0-4 m) are not affected by
contributions from these fluids and show predominantly seawater signatures. Despite the
elevated heat flow in the vicinity of the hydrothermal vent field, early-diagenetic reactions
are also not enhanced as seen e.g. by only slightly elevated $NH_4$ concentrations and sulfate
concentrations that remain at seawater values throughout the cores (Fig. 4).

498        4.2.2 Cold seeps


The selection of sampling sites at presumed seep locations was based on existing published
data (Lizarralde et al., 2010) and information from seismic records (see Fig. 2). Seismic
amplitude blanking along vertical zones below the seafloor indicates (active?) fluid conduits
at North and Central Seep. Following the hypothesis that sill intrusions and related high-
temperature alteration of sediments are driving the seepage, the expectation was to find
deeply-sourced (average sill depth ~400m) fluids, characterized by a typical geochemical
signature analogous to findings at Black Smoker vents in the Guaymas Basin (Von Damm et
al., 1985; Berndt et al., 2016). Such characteristics are e.g. a high concentration of
thermogenic hydrocarbon gases formed by organic-matter degradation, which is
accompanied by enrichments in other organic tracers such as ammonium as well as
depletion in Mg and a strong enrichment in fluid-mobile tracers like Li and B (e.g. Aloisi et al.,
2004; Scholz et al., 2009).
The results from samples obtained using a video-guided MUC show that the highest
methane concentrations compared to all other sites were measured at North, Central, and
Ring Seeps (Fig. 4b). This and the fact that methane concentrations are exceeding those at
the high-accumulation slope station underlines the visual evidence (abundant
chemosynthetic biological communities) of active methane seepage. At the two most active
sites, North and Central, high methane levels are accompanied by a significant drop in
sulfate and increase in alkalinity and $H_2S$, providing evidence for AOM. These pore water
trends are even more pronounced in GC01 (North) where the AOM zone was completely




penetrated and gas hydrate was found at about 2.5 mbsf. Unfortunately, GCs from similarly
active sites could not be obtained from Central and Ring seeps, mainly because of patchiness
of seepage spots and widespread occurrence of authigenic mineralizations at the seafloor
preventing sufficient penetration. Nevertheless, the occurrence of active methane seepage
at all three investigated sites is evident. A closer look at the lower panel of Fig. 4 a,b (and
Table S2) illustrates that the methane flux is not accompanied by any significant excursion of
major pore water constituents (e.g. Mg, Cl, Li) that would be typical for deeply-sourced,
high-temperature sediment-water interactions. Also Sr concentrations show seawater values
throughout all seep sites (Fig. 5), with the exception of North Seep where Sr concentrations
in conjunction with Ca (not shown) decrease and point to co-precipitation with Ca during
carbonate formation. The $^{87}Sr/^{86}Sr$ ratios show predominantly seawater signatures as well
(Fig. 5, Table S2). Similarly, $NH_4$ concentrations, as tracer for the intensity of organic matter
decomposition, in both MUCs and GCs, remain at levels <1mM. This is much lower than the
end-member reported from vent fluids in the Southern Trough (Von Damm, 1985) and also
lower compared to high-accumulation areas like the Slope and the Graben Site (Fig. 4a,b).
Essentially, all data presented in Figure 4 show that, with exception of methane and sulfate,
the pore water corresponds to ambient diagenetic conditions, typically met in this shallow
subsurface depth.  An explanation for the decoupling between high methane levels, sulfate
depletion at shallow depths, and otherwise more or less unchanged pore water composition
is that only methane in form of free gas is rising to the seafloor. This assumption requires a
closer look at the composition of dissolved hydrocarbons in general, which is given below.

4.3 Origin of hydrocarbon gases
4.3.1 Alteration effects

The origin of hydrocarbon gases can be deciphered by plotting hydrocarbon $CH_4/(C_2H_6+C_3H_8)$
ratios versus $\delta^{13}C_{CH4}$ data in a modified Bernard diagram (Schmidt et al., 2005 and literature
therein) (Fig. 8) and $\delta^{13}C_{CH4}$ versus $\delta D_{CH4}$ after Whiticar (1999) and Welhan (1988) (Fig. 9).
Most of the measured stable isotope data of pore water methane indicate a microbial origin
or a mixed microbial and thermogenic origin (Fig. 8, 9). By contrast, the isotopic and
geochemical signature of hydrocarbons venting at the Smoker Site reflects a mixture of



methane of thermogenic and abiogenic (methane derived from water-rock interactions)
origin (Berndt et al., 2016).

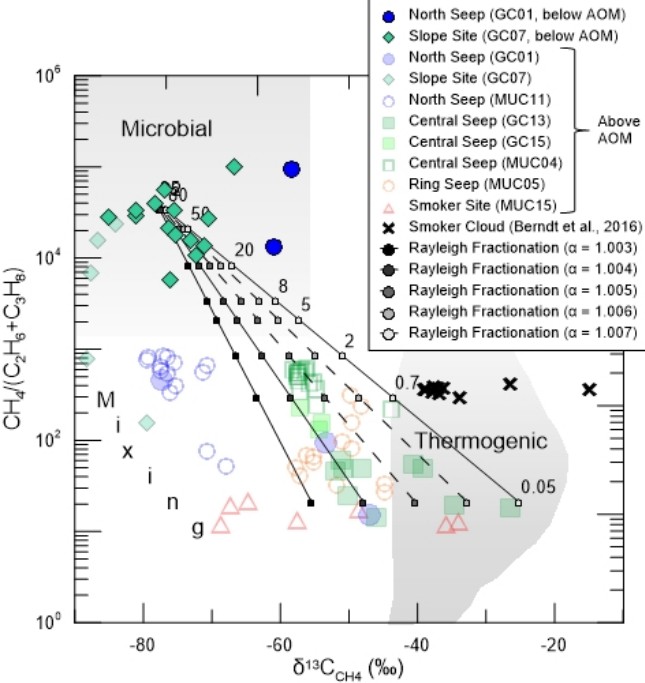


Figure 8: Hydrocarbon $CH_4/(C_2H_6+C_3H_8)$ ratios versus $\delta^{13}C_{CH4}$ data are shown after a modified
Bernard diagram (Schmidt et al., 2005). Pale symbols indicate samples above the AOM.
Rayleigh fractionation lines show the effect of (microbial) methane oxidation, labels indicate
the residual methane in %.





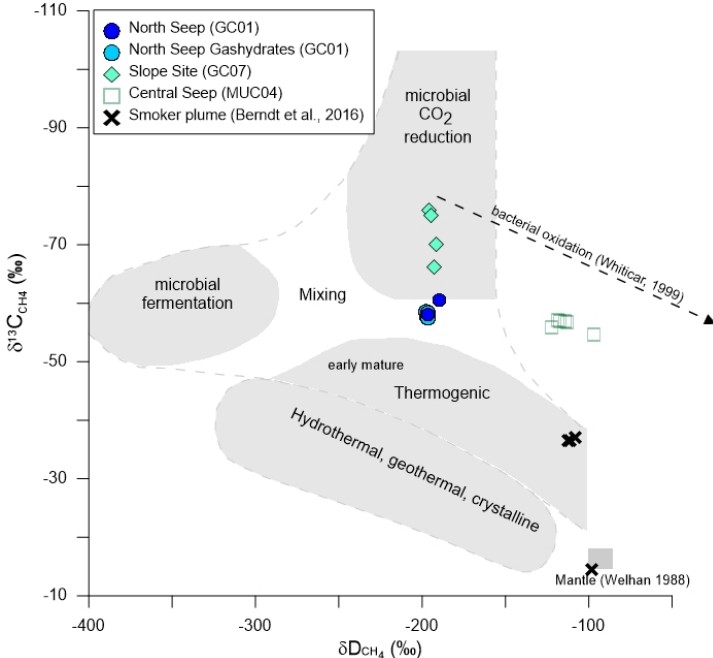


Figure 9: Carbon ($\delta^{13}C_{CH4}$) and hydrogen $\delta D$ isotope data after Whiticar (1999) and (Welhan,
1988). Pale symbols (Central Seep (MUC04)) indicate samples above AOM.

Interestingly, all but two samples from North Seep sediments are located above the AOM
(see Fig. 4) and could therefore be affected by oxidation (Fig. 8). Anaerobic methane
oxidation enriches $CO_2$ in $^{12}C$ which results in a progressively $^{13}C$-enriched methane residue
shifting the $\delta^{13}C_{CH4}$ values towards heavier values (e.g. Borowski et al., 1997; Dowell et al.,
2016). Considering Slope Site methane signatures as a microbial endmember composition
for the Guaymas Basin (Fig. 8), most of the data fall on calculated fractionation lines for
methane oxidation following a Rayleigh trend (Whiticar et al., 1999). Methane sampled close
to the Smoker Site (MUC15) is obviously also affected by anaerobic methane oxidation (Fig.
8). This process has recently been described by Dowell et al. (2016), who detected bacterial
and archaeal communities in hydrothermal sediments of the southern trough of the
Guaymas Basin, which were found to catalyze the oxidation of methane and higher
hydrocarbons and shift $\delta^{13}C_{CH4}$ values to heavier signatures.

Origin of methane and oxidation effects can further be identified in the $\delta^{13}C_{CH4}$ versus $\delta D_{CH4}$
plot after Whiticar (1999) and Welhan (1988) (Fig. 9). Slope Site samples plot in the field of



microbial $CO_2$ reduction while Smoker hydrothermal plume samples plot in the thermogenic
field, one sample of the Smoker Site even points to a mantle signature, and thus show clear
potential endmember isotope signatures. North Seep samples (pore fluids and gas hydrates)
plot in the mixing region while samples from Central Seep clearly shift away from the
microbial field and are considered to be affected by bacterial oxidation (Whiticar, 1999).
Considering only methane below the AOM as being unaltered, two North Seep samples and
the majority of the Slope Site samples show a clear microbial source of methane (Fig. 8). All
other samples appear to be affected by high degrees of oxidation following a Rayleigh
fractionation process and show that only a fraction between 2 % (MUC 04, Central Seep) and
0.05 % (GC15, Central Seep) remains as unoxidized methane (Fig. 8).

4.3.2 Origin of unaltered samples

Unaltered North Seep samples show a mixing origin in the $\delta^{13}C_{CH4}$ versus $\delta D_{CH4}$ plot (Fig. 9),
possibly stemming from microbial and thermogenic sources. Similar mixing signals of
thermogenic and microbial methane have also been observed at Hydrate Ridge (Milkov et
al., 2005) and seem to be a common phenomenon in hydrothermal and cold seep affected
sediments. In a few samples from North and Central Seep ethane concentrations have been
high enough to measure stable carbon isotopes and the $\delta^{13}C_{C2H6}$ values point to a
thermogenic origin of ethane (Table S3).

4.4 Timing of active (thermogenic) methane release

Based on the presented data set, even when considering some uncertainties with respect to
the fraction of thermogenic methane, the lack of any other geochemical evidence underlines
that probably no deep-sourced fluid is migrating upwards at present at the cold seepage
sites (compare deep-sourced seepage sites from the Gulf of Cadiz in Fig. 7). Hence, in terms
of the original hypothesis that fluid emanation is directly linked to recent sill intrusions, the
investigated "cold seep" sites cannot be considered as being active as claimed by Lizarralde
et al. (2010), who argue that thermogenic carbon is released up to 50 km away from the rift
axis causing a maximum carbon flux of 240 kt C yr$^{-1}$. First results by Lizarralde et al. (2010)
showed temperature anomalies, high methane concentrations, and helium isotopic
anomalies indicative of a magmatic source above bright features identified as bacterial mats,
tubeworms, and authigenic carbonate. These features are situated above areas of shallow
gas above sill intrusions comparable to structures identified in this study by seismic data (Fig.
2). The more detailed results of this study regarding pore fluid, water column, and gas
geochemistry show that only traces of thermogenic methane were found up to ~20 km off
axis (North Seep) and most methane was of microbial origin (Fig. 8, 9). Even pore fluids taken
close to the hydrothermal vent area are dominated by shallow microbial degradation
processes, indicating that hydrothermal fluid flow in the Guaymas Basin is rather localized
and bound to focused fluid pathways. The temperature and chemical anomalies detected by
Lizarralde et al. (2010) could also stem from the deep water layer in the Guaymas Basin itself
which is influenced by hydrothermal fluids (Campbell and Gieskes, 1984). Hydrothermal
activity in the Guaymas Basin produces hydrothermal plumes which rise to 100-300 m above
seafloor and then spread out along density gradients throughout the basin (Campbell and
Gieskes, 1984). Results of this study show that the Guaymas Basin has a well-mixed bottom
seawater layer consisting of patchy and elevated $CH_4$, as well as temperatures ranging
between 2.8 and 4.5°C in >1000 m depth (Fig. 6 and 10, Table S4). Off-axis methane
concentrations vary quite considerably and show e.g. a range from 6 to 28 nM for Ring Seep
and a temperature range from 2.8 to 3.9 for Central Seep. These bottom seawater
variabilities are bigger than the reported anomalies by Lizarralde et al. (2010) and indicate
that their findings might have been overrated.

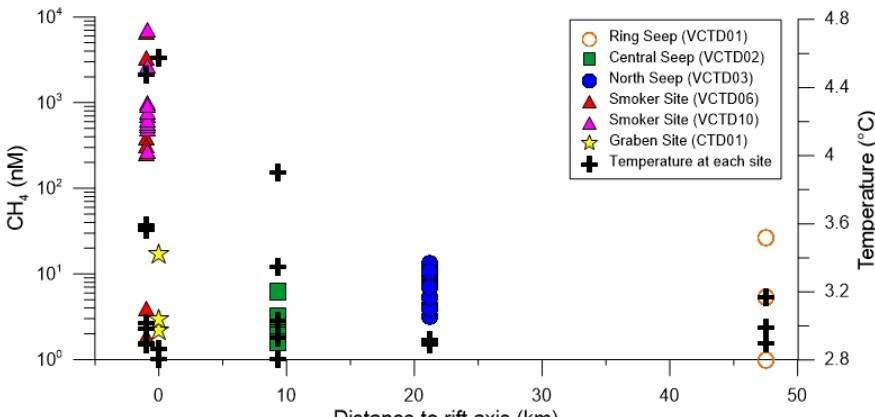




Fig. 10. Water column $CH_4$ (colored symbols) and temperature (black crosses) at cold seeps
and Smoker/ Graben sites relative to the rift axis.

Pore fluids taken in a transect from the rift axis up to ~20 km away show no evidence for
seepage of fluids that are affected by high-T reactions (Fig. 4). Shallow microbial degradation
processes determine pore fluid signatures and control the majority of the released methane
(Fig. 4, 8). It is likely the case that high temperature thermogenic reactions acted during sill
emplacement and released large amounts of carbon, but these processes appear to have
ceased since then. However, pipe structures still may act as high-permeability pathways and
facilitate the advection of gas. Small amounts of thermogenic carbon might still be released
as seen in microbial and thermogenic mixing signatures of $\delta^{13}C_{CH4}$ and thermogenic $\delta^{13}C_{C2H6}$
isotope data at North and Central Site. However, present methane advection rates are slow
(probably <1 cm $yr^{-1}$) as observed by low methane gradients in the pore fluid profiles (Fig. 4).
These conditions favor an effective turnover of $CH_4$ to bicarbonate and authigenic
carbonates by AOM (Karaca et al., 2010; Wallmann et al., 2006). The porous authigenic
carbonate block recovered from the seafloor at Central Seep can provide long-term
information about seepage in this area. The predominant biomarkers found in the seep
carbonate from the Central Site (56-VgHG-4) are consistent with an origin from dual species
microbial consortia perfoming the anaerobic oxidation of methane (AOM). High relative
abundances of crocetane and *sn*2-hydroxyarchaeol, along with DAGE, indicate major
contributions from methanotrophic archaea of the ANME-2 cluster and syntrophic sulfate-
reducing bacteria, probably of the *Desulfosarcina–Desulfococcus* group (Blumenberg et al.,
2004; Niemann and Elvert, 2008). These consortia appear to gain energy from AOM, with
sulfate as the final electron acceptor, according to the net reaction
$CH_4 + SO_4^{2-} \rightarrow HCO_3^- + HS^- + H_2O$
(e.g. Nauhaus et al., 2005; see Wegener et al., 2016 for a recent update).
The increase in alkalinity due to the AOM reaction plausibly explains the precipitation of
isotopically depleted authigenic carbonates. Particularly, ANME-2 biomarkers have been
reported in association with abundant fibrous, often botryoidal aragonite cements
(Leefmann et al., 2008), which is fully in line with the observations made at the Central Site
(see ch. 3.3). Moreover, the inferred major abundance of ANME-2 indicates that seep





carbonate formation once took place under high sulfate concentrations, strong  advective
methane flow, but no elevated  water temperatures (c.f. Nauhaus et al., 2005; Peckmann et
al., 2009; Timmers et al., 2015). The observation of minor amounts of typical water column
sterols also shows that these seep carbonates do not only carry their inherent AOM
signature, but also captured detritus from the surrounding sediment and background water
column sources during their ongoing cementation.
The bulk carbonate carbon isotope signature ($\delta^{13}C_{V\text{-}PDB}$ = -46.6‰) overlaps with the shallow
heavy $\delta^{13}C_{CH4}$ values (-27.5 and -48.6 ‰) in the pore fluids at Central Seep. Biomarkers found
in the bulk carbonate confirm a dominant AOM signature with a significant planktonic and
potentially $\delta^{13}C$ diluting background signal (Fig. S2). The oxygen isotope signature of the bulk
carbonate points to a low formation temperature of about 3°C. This is consistent with a
formation at ambient seawater which has bottom water temperatures between 2.8 and
3.0°C (Fig. 6, 10; Table S4). The $^{87}Sr/^{86}Sr$ analyses support this assumption by values within
uncertainty identical to modern seawater. Also U-Th carbonate dating performed at these
authigenic carbonates provide formation ages younger than 240 yrs BP. In conclusion,
authigenic carbonate shows a recent to sub-recent formation age with methane from
shallow sources at ambient seawater and thus confirms the results from pore fluid and gas
geochemistry of cessation of deep fluid and gas mobilization.
Taking a closer look at the seismic lines across the seep locations, it becomes obvious that
the disrupted sediment layers are not reaching to the sediment surface (Fig. 2a, c). This
implies that fluid mobilization ceased at some time before the uppermost sediment layers
were deposited. The doming above the Central Seep provides some clues on the timing of
fluid migration (Fig. 2c). Assuming that the doming is the result of buoyancy-related uplift
(Koch et al., 2015) it represents the time when intrusion-related gas reached the sea floor.
Assuming further a sedimentation rate of 1.7 m per 1000 years (Central Seep; Table 1) and
maxima and minima deposition depths of 48 and 12 m respectively below seafloor (see Fig.
2c) this would imply that most of the gas reached the seafloor between 28 and 7 kyrs ago.
Even assuming minima and maxima sedimentation rates of 3.5 m (North Seep) and 0.5 m
(Ring Seep) per 1000 years gas flow would have ceased at the earliest between 14 and 3 kyrs
ago or at the latest 96 and 24 kyrs ago. This finding supports the results of the pore fluid and





gas geochemistry which show no sign of active fluid flow from depth at cold seep sites in the
northern Guaymas Basin.
Large amounts of $CH_4$ (and $CO_2$) must have been emitted to bottom waters during the
calculated periods (s.a.), rapidly after sills intruded into the organic-rich sediments in the
Guaymas Basin. However, these carbon emissions must have ceased after sill-emplacement
ended and the impact on climate appears to depend on the durability of the magmatic
system.

**5 Conclusions**

Magmatic intrusions into organic-rich sediments can potentially release large amounts of
carbon into the water column and atmosphere and are therefore discussed as potential
trigger mechanisms for rapid climate change, e.g. during the PETM. In the Guaymas Basin,
off-axis cold seeps do not show indications for present-day hydrothermal activity. Pore fluids
sampled from cold seep structures and in the vicinity of hydrothermal vents in the northern
Guaymas Basin, are dominated by seawater concentrations and show no sign of deep fluids
or temperature-related diagenesis. Methane measured at the investigated sites stems from
a mixed origin (microbial and thermogenic sources), though mainly from microbial
processes. This may suggest that hydrothermal circulation has stopped at depth and, based
on seismic data, ceased more than 7kyrs ago. Sill-induced hydrothermal systems appear to
be an effective way to release carbon, but the period of time depends on the longevity of
the magmatic system.

**Acknowledgments**

This work was undertaken within the MAKS project which was funded by the German
Ministry of Science and Education (BMBF). We would like to thank the master and crew of
the R/V Sonne for their help and support during the SO241 cruise. Further thanks goes to
Regina Surberg, Bettina Domeyer, and Anke Bleyer for analytical support during the cruise
and on shore. We greatly appreciate the support from Ana Kolevica, Tyler Goepfert,
Sebastian Fessler, Andrea Bodenbinder, Yan Shen, and Jutta Heinze for onshore analyses.





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





Table 1: Station list and site names of GCs and MUCs taken in the Guaymas Basin with according water depth. Heat flow and temperature gradient data measured either attached to GCs or to a sediment probe.

| Site | Site name | Latitude (N) | Longitude (W) | Water depth (m) | Temp. gradient (K/m) | Heat flow (mW/m$^2$) | Sed. rate (m/kyr) | Mass acc. rate (g/cm$^2$/yr) |
|---|---|---|---|---|---|---|---|---|
| **Gravity corer** | | | | | | | | |
| St.07 - GC01 | North Seep Reference | 27°33.301' | 111°32.882' | 1845 | 0.14 | 28 | - | - |
| St.10 - GC04 | Site Central | 27°26.531' | 111°29.928' | 1846 | 0.22 | 140 | - | - |
| St.09 - GC03 | Seep Central | 27°28.138' | 111°28.420' | 1837 | - | - | - | - |
| St.09 - GC13 | Seep Central | 27°28.193' | 111°28.365' | 1838 | 0.16 | 113 | - | - |
| St.72 - GC15 | Seep Smoker | 27°28.178' | 111°28.396' | 1837 | - | - | - | - |
| St.51 - GC09 | Site Smoker | 27°24.472' | 111°23.377' | 1840 | 11 | 8069 | - | - |
| St.58 - GC10 | Site | 27°24.478' | 111°23.377' | 1845 | 10 | 6509 | - | - |
| St.47 - GC07 | Slope Site | 27°24.412' | 111°13.649' | 671 | - | - | - | - |
| **Multicorer** | | | | | | | | |
| St.33 - MUC11 | North Seep | 27° 33.301' | 111° 32.883' | 1855 | - | - | 1.7* | 0.05* |
| | | | | | | | 3.5# | 0.15# |
| St.23 - MUC05 | Ring Seep Reference | 27° 30.282' | 111° 40.770' | 1726 | - | - | 0.5 | 0.01 |
| St.15 - MUC02 | Site Central | 27°26.925' | 111°29.926' | 1845 | - | - | 2.3 | 0.04 |
| St.22 - MUC04 | Seep Smoker | 27° 28.165' | 111° 28.347' | 1839 | - | - | 1.7 | 0.04 |
| St.65 - MUC15 | Site Smoker | 27° 24.577' | 111° 23.265' | 1846 | - | - | 1.8 | 0.05 |
| St.66 - MUC16 | Site | 27° 24.577' | 111° 23.265' | 1842 | - | - | 2.1' | 0.08' |
| | | | | | | | 0.4+ | 0.02+ |
| St29 - MUC09 | Slope Site | 27°42.410' | 111°13.656' | 665 | - | - | - | - |
| **HF lance** | | | | | | | | |
| St.60a - HF008_P03 | | 27°24.273' | 111°23.396' | 1840 | 4.6 | 3206 | - | - |
| St.60a - HF008_P01 | | 27°24.623' | 111°23.626' | 1834 | 0.86 | 599 | - | - |
| St.60a - HF008_P02 | Smoker Site | 27°24.554' | 111°23.512' | 1840 | 2.8 | 1953 | - | - |
| St.60a - HF008_P04 | | 27°24.408' | 111°23.288' | 1849 | 2039 | 1427 | - | - |
| St.60a - HF008_P05 | | 27°24.341' | 111°23.177' | 1852 | 1014 | 710 | - | - |
| St.60a - HF008_P06 | | 27°24.265' | 111°23.082' | 1844 | 0.74 | 516 | - | - |



| | | | | | | | | |
|---|---|---|---|---|---|---|---|---|
| St.60b - HF008_P07 | | 27°24.193' | 111°23.956' | 1834 | 0.8 | 579 | - | - |
| St.60b - HF009_P04 | | 27°24.543' | 111°23.351' | 1837 | 15 | 10835 | - | - |
| St.60b - HF009_P01 | | 27°24.605' | 111°23.317' | 1837 | 0.39 | 274 | - | - |
| St.60b - HF009_P02 | | 27°24.552' | 111°23.347' | 1834 | 3451 | 2415 | - | - |
| St.70 - HF011_P01 | | 27°25.802' | 111°25.486' | 1870 | 0.38 | 262 | - | - |
| St.70 - HF011_P02 | Graben Site | 27°25.460' | 111°24.946' | 2019 | 0.48 | 338 | - | - |
| St.70 - HF011_P03 | | 27°25.955' | 111°24.493' | 2046 | 0.43 | 302 | - | - |
| St.70 - HF011_P04 | | 27°25.837' | 111°24.951' | 2025 | 0.46 | 320 | - | - |
| **Authigenic carbonate** | | | | | | | | |
| St.56- VgHG-4 | Central Seep | 27°28.181' | 111°28.379' | 1843 | - | - | - | - |

*·#Sedimentation and mass accumulation rates at Station 33 of the 0-13 cm, 13-18 cm layers, respectively

'·+Sedimentation and mass accumulation rates at Station 65 of the 0 - 7 cm, 7 - 17 cm layers, respectively