# Peer review of "On the formation of hydrothermal vents and cold seeps in the"

_Biogeosciences, 2018_

## Referee Comment (RC1) · Anonymous Referee #1 · 11 Mar 2018

In this paper, the authors presented a rich set of geochemical data from pore fluid, water column, and authigenic carbonates to argue against the hypothesis by Lizarralde et al. (2010) that, hydrothermal intrusion in the Guaymas Basin recently induces methane seepages by producing methane from thermally decompose organic matter. The authors argue that the seepage, if it ever occurred, must have ceased several thousand years ago First of all, I would urge the authors to check the manuscript more carefully, there are numerous places with very obvious grammatical errors, unexplained abbreviations, and incorrect reference to the figures. Also, some of the long sentences and excessive use of comma make it difficult to read sometimes. I tried to point out some of these and hope the authors consider these comments as friendly critics from

a reader's point-of-view. In terms of the interpretation and conclusion, I do not disagree with what the authors proposed but however do not think their interpretation is the sole explanation of their observations. Their primary observations are: 1) porewater geochemistry from the Smoker Sites is dominated by seawater signal with a minor contribution of hydrothermal fluid; 2) from the seep sites, porewater geochemistry show no sign of deep-sourced fluid, despite the high input of methane from a mixture source of biogenic and thermogenic methane; 3) the age of authigenic carbonate is dated to be very young with geochemical signatures all indicate a close-to-modern seafloor condition. The (1) observation was interpret by the authors as a seawater circulation in shallow sub-surface that has been observed in other hydrothermal systems. The (2) observation was interpret by the authors as due to the decoupling of gas from the water phase. With all these observations, the author concluded that the seepage has stopped now and therefore the mechanism proposed by Lizarralde et al. (2010) is over-rated. With the same set of observations, one can also interpret that the shallow circulation seen in the porewater profiles is driven by a recent hydrothermal activity that provides the heat source (as shown by all the temperature measurements presented in the paper) to decompose organic matter and therefore explains the mostly biogenic source of methane observed from the cold seep sites. The decoupling of gas and water phases is not uncommon and does not exclude the contribution of deep water; it might arrive later than the gas phase or at a different location. Also, the circulation of seawater must have diluted the signal from deep-sub surface. The young age dated from the authigenic carbonate also support such recent seepage event. The above interpretation turns the same set of observation to support the hypothesis proposed by Lizarralde et al. (2010). I want to emphasize it is difficult to interpret the observations of "no anomaly". The authors must be more careful about this.

I also think the title of the paper misleading. From the title alone, it appears that they are in favor of the transition from hydrothermal vents to cold seeps. I also think the authors should present their opinion already in the final sentences in the introduction (e.g., Line 97-99). When I read this, I thought they agree with the transition from

hydrothermal vent to cold seep until later in the discussion.

Specific comments:

Line 24: What does the 500m here mean?

Line 31: If pore fluid is predominately seawater than you wouldn't call it "cold seep pore fluid".

Line 48: Kennett 2000 is not a good citation as this paper only dealt with the Quaternary excursions not PETM.

Line 74: delete "that"

Line 89: "a helium isotope signature indicative of mid-ocean ridge basalt." I guess you mean He isotopic signature tells them the fluid came from mid ocean ridge.

Line 90: up to several hundred of meters.

Line 91: "magmatic intrusions into underlying sediments" The orientation is weird in this sentence. Here the underlying should refer to the magmatic intrusion. Do you mean the intrusion penetrated strata deeper than it was?

Line: 94-97: Could you check the sentence again? If you intent to use two commas to form a clause, please remember to close the clause by adding the second comma. Also, consider using an active tone in this sentence, such as " during the SO241, we sampled at XXX and XXX locations."

Line 105: were

Line 106: check the articles of this sentence, not always "a"

Line 109: locations of seeps

Line 115: why need "respectively" here? What is GI gun?

Line 126: I assume you mean authigenic carbonate concretions
Line 127, 128: "Hence, comparing results from different seeps might be biased in this regard." Unclear what you mean.

Line 131-134: The way you use comma is really confusing. For example, "at three seepage sites, North (GC01, MUC11),Central (GC03, GC13, GC15, MUC04), and Ring Seeps (MUC05)," Do you intent to say the three seepage sites include north, central and ring seeps sites? or the "three seepage site" is another site other than north, south, and Ring Seeps.

Line 133: Are you sure you gave definition of the reference site "above" not "below"?

Line 155: "at a sampling rate of 1s." sampling rate of what?

Line 171: "were" Line 187-193: I understand one can sure find details in the paper cited. However, I think it's important to mention things that are absolutely crucial. For example, it is important to mention how soon were the HS and ammonium analyzed after recovery of the porewater as both species are easily degraded due to oxidation and microbial consumption. It's also known that ammonium measurements by photometry method are heavily impacted by the presence of HS. What treatment did you do to prevent that. Titration of alkalinity is also a time-sensitive analyses as carbonate precipitation is still happening in the water samples. For the cation and anion samples brought back to shore lab, what preservation measure was performed. All of such information are crucial and I would like to see more description in the main text but not just "please refer to XXX".

Line 194-198: As volcanic material might be present in the study area, it is important to check the abundance of Rb and see if that affect the strontium isotopic ratios. This is supposed to be a routine for analyses like this. I would like to see some more information on this.

Line 209: VPDB needs to be explained

Line 234: where in the supplement?

Line 231-241: This appears to be a ridiculously long sentence. Please revise the whole paragraph so that it's more readable.

Line 272: blankening? Blanking?

Line 272-273: Im not a geophysicist but I thought the blanking zone in seismic profile is due to gas/water (stuff with low density) instead of sediment mobilization?

Line 287-290: check the unit for 60, 15 mbsf. I think you mean ms. Also, explain what is mbsf.

Line 317: what do you "lower meter"? do you mean shallow in GCs?

Line 333: photometry method measures total hydrogen sulfide, S2-, HS-, and H2S. Please revise throughout the text. Why for some ions you specified their charge (like SO42-) for others you ignored the charge (NH4, Li, Mg)? Also, please revise alkalinity to total alkalinity (TA) for clarity throughout the text.

Figure 4: From the figure, the TA from GC07 could be as high as over 70 meg/L however the highest value listed in supplementary is only 65 meg/L. Could you check this again? Also, I suggest modify the scale of the plot. For example, it is really hard to see the changes in Mg and Li concentrations from the plot despite the 10% increase and decrease in concentrations of these two ions. The figure should be able to reflect these variations better.

Line 347: revise to TA and total HS.

Line 350: I do not agree Mg and Li concentrations are similar to seawater for all sites, you apparent have higher Mg and Li in GC07

Line 370: the lowest and highest values I can see are -26.5 and -88.2.

Line 373: I don't see any dD-CH4 value reported for Smoker unless you mean VCTD data, which is not from porewater.

Line 385: There is no VCTD09 in your data from supplementary and figure 6.

Line 398: I wonder what kind of calcite it is, high-Mg or low-Mg calcite.

Line 403, 403: isn't the reproducibility should be reported in the method section.

Line 421: I wouldn't be so sure about this conclusion. Besides of methane from thermogenic degradation of organic matter, it is possible you have methane from hydrothermal activity, which is not much related to the organic matter. This would make sense with the mantle source helium reported in Berndt et al. (2016). I also suggest you report the exact value of helium isotopic anomaly reported by Berndt et al here, so readers could have a better sense of the information.

Line 425: Check the format of citation.

Line 434-435: If you look closely to the raw data, both Sr and Ca concentrations are 10% elevated compared to the seawater value. Also one of the only two 87Sr/86Sr values reported from GC09 shows significantly lower value from seawater values. This again emphasize the authors should really adjust the scale of the plot (Figure 4) to reflect these small but significant changes.

Line 445-446: again, if you look into the data clearly you would probably slightly change the conclusion here.

Line 449-450: You only have one indicator, NH4, reported here. I don't think you can justify for all. Not to mention NH4 concentration is affected not only by organic matter degradation but also cation exchange.

Line 455. I don't see why is relevant to refer fig 3 here. I thought you mean fig4. Also, this sentence is so odd. I don't quite sure I get your point. How do you know it's high level of AOM but not just sulfate reduction+organic matter degradation, which is in line with your high TA and NH4 levels.

Line 459-460: Of course the data could be explained this way, but alternatively, if there

is just no input of methane from the Smoker site (GC09, GC10), then one would expect exactly the same porewater profiles as reported here. The present data provide no justification of whether seawater convection exists or not at these coring sites.

Line 461: now you mentioned the Li anomaly. I think this observation should be mentioned earlier in the text.

Line 464-466: both Sr and Ca concentrations are also slightly elevated and the one $^{87}Sr/^{86}Sr$ value from GC09 is also significantly lower than seawater value.

Line 464: What is the cause of high Li? Hydrothermal solution (Line 465) or mineral composition (Line 462). If the authors think it's the latter, you should provide a explanation of the process and how.

Line 473-474: I in general agree this conclusion but think this paragraph could be better integrated with the paragraph discussing the porewater data of Smoker site. Especially the statement here is in contradiction to the statement in Line 434-435.

Fig. 7: what is the x-axis of (A)? Also, how the mixing lines were determined in (a) and (b), especially in the log-log plot and log-linear plot. I think for the mixing lines should look differently the ones from the current plots.

Line 491-494: The authors really need to work on this statement to get a self-consistent conclusion on this. See my earlier comments on this.

Line 502 "(active?)" appears without context. Please clarify.

Line 505-509: Since methane can also be generated through hydrothermal activity and even abiogenic processes, I don't see why organic matter degradation signal is necessarily expected.

Line 526: In my view, it's weird to see one calls Li as a major porewater constituent, as it's only less than 30 microM in the porewater.

Line 531 as a tracer

Line 565: what kind of oxidation of methane you are talking about? Aerobic or anaerobic?

Line 566: AOM enriches DIC in 12C.

Line 566-568: Im not sure how you this process you described can help explain your data. Besides, if you look into the Borowski et al (1997) paper, the paper is intent to explain why d13C-CH4 is actually counter-intuitively light in the AOM zone. It's true that AOM supposes to make the residual methane heavier in isotopic signature but this is not what usually observed and definitely not what Borowski et al intent to explain in their paper.

Line 570: for anaerobic methane oxidation. It's important to specify which oxidation.

Line 601-629: In the argument against the conclusion by Lizarralde et al., how does the observation the authors had, a convection of seawater into the shallow sub-surface in the Smoker Sites, affect such argument. It is likely that seawater convection in the hydrothermal is a short-term and contemporary process, the geochemical signal happened to be capture by the current study. In this case, how do you actually use the observation of no geochemical signal to argue against the conclusion by Lizarralde et al. Besides, the convection of seawater in hydrothermal regions must be driven by seeping of fluid in the hydrothermal vents. If as the authors claimed, the porewater profiles are indicative to seawater convection, isn't that just confirmed the hydrothermal activity?

Line 656: It's unexpected to see the authors show AOM reaction such late in the paper as they have talked about a lot earlier in the text. I suggest move part of this discussion when they use porewater profiles to infer intensive AOM activity.

Line 677-680: I agree that the various lines of evidence from the carbonate suggest the recent formation but I don't see how do these support the conclusion "cessation of deep fluid and gas mobilization" the authors derived from porewater data. Isnt that the

young ages from authigenic carbonate suggest a very recent seepage event? Since porewater profiles are probably contemporary signatures, can really conclude that the seepage has died just because they see nothing from the porewater profiles? Similar to my earlier comment, the "boring" and seawater-like porewater profiles were interpreted by the authors as due to seawater convection in the shallow subsurface. If this is true, how can the authors use this to say that the deep fluid migration has stopped?

Line 696: what is s.a.

Supplement tables:

Please revise the units of mmol or micromol to mM and microM throughout the table. There is no such unit.

The meaning of "-" in all the tables are unclear. Does it mean samples/analyses are not available or it is below detection limit. Especially for the table of d13C and dD of methane, not clear why sometimes there is not measurements of dD despite the high concentration. Also, it's not clear how "-" different from just a blank in the table.

---

## Referee Comment (RC2) · Anonymous Referee #2 · 1 May 2018

The manuscript bg-2018-12 by Geilert et al. reports geophysical and geochemical observation at the Guaymas Basin. Authors collected seismic dataset and porewater/seawater geochemistry at/above the hydrothermal vent and cold seep sites and interpreted them to discuss temporal transition of the magmatic activity.

I do not recommend the manuscript to be published on BG because of three major concerns. First, spatio-temporal scales are mismatched between geophysical and geochemical datasets collected. Particularly, spatial coverage of geochemical profiles collected is quite limited to discuss basin-scale phenomena suggested by seismic observation. Such limited evidence allows frail interpretation to reveal general characteristics

of the seep and vent sites. Second, construction of the manuscript is complicated, and I cannot catch the main story. Separation of what this study actually achieved from the achievements in previous studies (Lizarralde+ 2010 and Berndt+ 2016) is unclear. Third, as I read, main focus of this study is not biogeoscience. It seems geological and/or geochemical study because major purpose of this study is to characterize subseafloor geology/geochemistry and to bridge them. Although microbial activities of methane production/consumption are discussed, it is not main story of this study. Specific comments are presented below.

L001: I think more specific wording describing what authors observed seems better.

L021: This sentence seems inadequate as abstract of this study.

L024: In a research field for hydrothermal activity, horizontal distance of $\sim$500m is not "close". See Cruse&Seewald 2006; 2010; Reeves et al. 2011; Baumberger et al. 2016; or some other numerous papers.

L040: Introduction, carbon flux from seafloor to atmosphere, is not closely related to what authors observed in THIS study.

L051: This paragraph can move to M&M.

L069: What is environmental conditions?

L071: Magmatic intrusion is geological process while fluid-rock and fluid-sediment interactions (associated with magmatic heat) influences fluid/sediment geochemistry. Because major part of this study is geochemical description, it seems better to make the wordings clear.

L075: These sentences (L075-082) seem inadequate for this study.

L097: Authors do not clearly state whether seismic data is acquired in this study or not. Clarify it.

L125: Microbial mat is adequate.

L131: I feel the names of samples seem confusing. Rename of the samples based on geological or geochemical properties, such as North Seep site samples (NS01, NS02, NS03) and smoker site (SM01), seems better for readers.

L137: immediately "subsampled"

L139: Please show a reference for pressure filtration.

L141: What is difference from core retrieval in L137?

L144: Please show a reference for centrifugation.

L150: Purpose of temperature and conductivity measurements is unclear.

L167: Names seem confusing.

L208: MAT 253?

L217: What was the sample analyzed?

L255: Purpose of biomarker measurements is unclear.

L292: Please show (raw) vertical profiles of temperature in addition to (processed) heat flow values in figure 3 or figure 4.

L384: Is the water column chemistry already reported in Berndt et al. 2016? Is it first reported in this study? Please clarify it.

L415: 4.2? 4.1?

L418: Is it from Berndt et al. 2016?

L425: Because horizontal distribution of heat flows are highly heterogeneous at around high-temperature vents, such comparison may make no sense.

L446: Is this hypothesis supported by previous observations at sedimented hydrothermal vent sites?

L455: Fig4?

L491: I guess chemical reactions between sediment and intruded sill occur only at the time of eruption event. Fluid-sediment interaction associated with magmatic heat source occurs more likely. See Cruse&Seewald 2006 GCA, Ishibashi et al. 2014 Geochem.J, or some other papers reporting fluid geochemistry of sediment-covered vent sites.

L599: The story of timing of methane release seems frail due to limited evidences for temporal scaling. Information about time is only derived from solid phase (carbonate geochronology and sedimentation rate), and no evidence about past methane release is presented. Although past intrusion into sediment suggested by seismic dataset may imply generation and and release of thermogenic methane at the time of intrusion, it is just interpretation.

L703: This is not conclusion of this study.

L712: This interpretation has been clear before this study and is not proved in this study.

Fig1: Not informative. Except DSDP site and zoom up for seep-vent sites are better.

Fig3: Y-axis scaling is not good. Using two panels for large and small heat flows is better.

---

## Author Comment (AC1) · 16 May 2018

The referee criticizes in his comment mostly our conclusion that hydrothermal activity has ceased in the Guaymas Basin deduced from the observation of dominantly seawater signatures in the pore fluids and biogenic methane emissions. The referee proposes that our data could also be interpreted in an opposite direction, namely that recent hydrothermal activity drives a shallow convection cell that draws seawater into the sediment.

At first, we do agree with Lizarralde et al. (2010) that hydrothermal activity in the Guaymas Basin was once driving seepage and an elevated thermogenic methane flux to

the water column. We can also not exclude that there is still thermogenic methane re-leased into the basin driven by off-axis sills. However, as all seep sites investigated in this study show predominantly seawater composition, a simple correlation of detected sills and active thermogenic methane release as done by Lizarralde et al. (2010) ap-pears not to be feasible. Our data set shows that deep processes are extinct, at least at the investigated sites so that it is not unlikely that at other places deep processes are extinct as well. At least, it does not seem valid to assume that all other off-axis sills rep-resent active hydrothermal systems. In order to calculate accurate thermogenic carbon fluxes, sill emplacement mechanisms like longevity and spatial distribution need further investigation. It appears that this aspect is not well understood from our manuscript and will be edited if it is accepted for further revisions.

The referee claims that the heat provided by the Black Smoker field might 'decompose the organic matter and therefore explains the mostly biogenic methane source'.

However, if organic matter is decomposed by an elevated heat source than the isotopic signal would be indicative of this thermogenic source. The thermogenic $\delta$13CCH4 signal is relatively heavy (about -40 to -20‰ compared to the biogenic signal (< -55‰. All our (unaltered) $\delta$13CCH4 data falls in the biogenic field (see Fig. 8) and are thus not decomposed by thermogenic alteration. The lateral heat from the Smoker field might support and enhance biological processes but is not responsible for the isotopic signal as proposed by the referee.

The referee further criticizes that we just might not have detected the deep fluid phase as it is decoupled from the gas phase and might arrive later or at a different location.

We do not think that this is a likely alternative to the presented hypothesis. The known active hydrothermal systems from the Southern (Von Damm et al., 1985) and Northern (Berndt et al., 2016) rift axis in the Guaymas Basin emit hot fluids with clear evidence for high temperature fluid-rock interactions and thermogenic gas production. Such a fluid is not found at any of the seeps. Instead, we found pore water containing predominantly biogenic methane, but which is otherwise only slightly diagenetically altered from seawater. Biogenic methane formation is expected to occur within the uppermost tens to a few hundreds of meters below the seafloor at low temperatures. Methane-enriched pore water sourced in those depths should be likewise enriched in other products of organic matter degradation (e.g. NH4). Such a fluid composition was found at the Slope Site, where organic matter turnover is extremely accelerated by high accumulation rates of organic material. The fact that only biogenic methane is significantly enriched at the seep locations lets us conclude that methane gas is percolating through shallow sediments (even forming gas hydrates as observed at North Site) rising along pre-existing low permeability pathways formed by previous hydrothermal activity.

We argue that such a high-temperature geochemical signal might have been present during and for a certain time after sill emplacement, but as there are no obvious pore water signatures today, we conclude that deep processes are extinct at the investigated sites (see also section 4.4). The young age of the carbonate supports our hypothesis of a decoupled gas and fluid phase as only ascending gas is needed to drive AOM and the formation of authigenic carbonates. Biomarker, $\delta 13CCH4$, $\delta 18OCaCO3$ and 87Sr/86Sr signatures clearly point to a formation in seawater at ambient temperatures. We agree with the referee of a recent seepage event, however, mainly driven by shallow-sources, biogenic gas and not by deep-sourced, hydrothermal processes.

Overall, we do agree with Lizarralde et al. (2010) that during and for a certain time after sill emplacement large amounts of CH4 and CO2 are emitted as the heat released by magmatic intrusions induced thermogenic decomposition of organic matter. However the activity of such a process appears to be limited by the lifetime of a sill-induced hydrothermal system (time required to cool down). From sediment thicknesses above extinct fluid conduits we estimated that the processes must have stopped more than 7kyrs ago at least at the places investigated so far. We cannot exclude that there are still areas in the Guaymas Basin with active sill-induced methane release. At our investigated sites though, we have not found any evidence of thermogenic methane

release. A simple extrapolation as done by Lizarralde et al. (2010) in which they compile all sills and estimate the potential methane release appears not applicable.

The referee claims that the title of our manuscript might be misleading as it appears to argue for the process of active thermogenic methane release.

The title of the manuscript might indeed be misleading. We referred to the processes itself and not to the activity. We are willing to change the title to a more general meaning, like 'On the development of hydrothermal vents and cold seeps in the Guaymas Basin, Gulf of California'.

Specific comments:

We also appreciated the helpful and detailed specific comments about language, grammar, and general clarification needs, which we will gladly correct in case of a positive evaluation. In L434-435, L445-446, L461, and L473-474 in the referee comment the referee criticizes our interpretation of seawater and hydrothermal signals in the pore fluids.

We did find hydrothermal signatures but only in the deep core section of GC09 (>4m) as reported in lines 461-474 in the manuscript. The remaining pore fluids show seawater composition and are interpreted as shallow convection cell drawing seawater into the sediment. We propose to rearrange this section to clarify our interpretation of the data.

In L502 the referee notes that the word 'active?' appears without context.

The blanking of the seismic profile indicates a fluid conduit, but the profile cannot differentiate between active fluid and / or gas flow. As we only conclude later in this paragraph that the fluid and gas phases must have been decoupled we decided to put the 'active' in question.

In L526 the referee criticizes that Li is named a major pore water component.

Li is considered as a major indicator for high-temperature sediment-water interactions,

as are Mg and Cl. We agree that it might be confusing in this context and are willing to rearrange the sentence.

In L565 and following the referee requests a definition for the type of oxidation.

The samples were taken in the AOM zone and the oxidation is therefore anaerobic. We will gladly clarify this section during the revision process.

In L601-629 the referee claims that the shallow convection cell surrounding the hydrothermal area contradicts with our conclusion.

We do not deny the activity of the hydrothermal system in general. We just state that at the investigated seep sites no deep signal is detected and there are no indications of actively released thermogenic methane. We cannot exclude that this process occurs in other areas of the basin, however Lizarralde et al. (2010) calculated methane flux might be excessive (see also comments above).

In L677-680 the referee wonders how the young age of the carbonate argues for a cessation of a deep signal.

The $\delta$13CCH4 data of the bulk carbonate overlaps with the $\delta$13CCH4 values in the associated pore fluids. No indicators of a deep signal have been found in the carbonate. Indeed, carbonate formation requires a recent seepage event; therefore we concluded beneath others that the fluid and gas phase must have been decoupled (see also comment above).

References

Von Damm, K. L., Edmond, J. M., Measures, C. I. and Grant, B.: Chemistry of submarine hydrothermal solutions at Guaymas Basin, Gulf of California, Geochim. Cosmochim. Acta, 49(11), 2221–2237, 1985.

Lizarralde, D., Soule, S. A., Seewald, J. S. and Proskurowski, G.: Carbon release by off-axis magmatism in a young sedimented spreading centre, Nat. Geosci., 4(1), 50–

54, doi:10.1038/ngeo1006, 2010.

---

## Author Comment (AC2) · 16 May 2018

Referee #2 has difficulties with three main aspects of our manuscript concerning the biological significance, the spatial coverage of sampling sites, and the new discoveries of our manuscript in contrast to earlier studies.

First, the referee claims that the biological aspects of our study are too small to get published in Biogeosciences.

The main findings and conclusions of our study are based on biological aspects, like the microbial signature of $\delta13C$ data and the AOM-dominating biomarkers identified in the

carbonate. The detected microbial signatures helped to identify that deep processes are extinct nowadays. The biological results support our geochemical and geophysical observations and form a key point of our discussion. Additionally, we understand that the objective of this journal is to publish research which combines biological, chemical, and physical investigations and which highlights the interaction between them (see homepage Biogeosciences). Our manuscript combines all three aspects and emphasizes the importance of an interdisciplinary research approach to draw the best possible conclusions.

Secondly, the referee expresses his concerns that the spatial coverage of our sampling sites is not sufficient to infer basin-wide phenomena.

Sample locations were chosen based on findings by Lizarralde et al. (2010) who describes sill intrusions associated with hydrocarbon gas emissions, biological communities, and authigenic carbonates. In this study, we investigated 3 seepage sites at various distances from the hydrothermal vent field based on locations identified by Lizarralde et al. (2010) as areas of active methane release. Additionally, a reference site, smoker sites as well as the water column have been sampled. With the exception of the active smoker site, there is no indication for a deep fluid advection and methane $\delta$13C data are predominantly of microbial origin (see Fig. 4 and section 4.4). Despite the fact that no deep fluids were detected at the seepage sites, an active methane flux was present, indicating that we hit the currently active sites described in Lizarralde et al. (2010). The detected methane was predominantly of microbial origin and no active thermogenic methane is released nowadays at the investigated sites as claimed by Lizarralde et al. (2010). We cannot exclude the possibility that thermogenic methane is still released in other areas of the basin, but the lack of evidence for high temperature geochemical processes at the investigated sites contradicts with Lizarralde's et al. (2010) conclusions. The seismic evidence of seep-induced hydrothermal systems alone is not sufficient for projecting methane emissions for the whole basin at present (see also comment to referee#1). Thermogenic methane release induced by off-axis

sill intrusions is still a likely process to occur, but our study suggests that the lifetime of these systems is limited and has to be taken into account for budget calculations. Hence, the study of Lizarralde remains valid and is highly valuable in terms of describing the general process and the potential magnitude, but care has to be taken concerning the longevity of the hydrothermal systems and associated thermogenic methane release after the occurrence of sill intrusions.

The last point of criticism by the referee is that it is not clear how the findings of this study differ from those of Lizarralde et al. (2010) and Berndt et al. (2016).

If the main findings of this study are not clear to the referee then we indeed have to improve this part of the manuscript and clarify how our findings differ from those of earlier studies. While Berndt et al. (2016) focused on characterizing the geophysical and geochemical characteristics of the Smoker area, Lizarralde et al. (2010) investigated geophysical aspects of the wider basin and the water column. Our study is the first one to look at geochemical, biological, and geophysical characteristics of seepage sites and the water column above. Main findings are the decoupling of gas and fluid phases, the microbial origin of methane, and the detection of sediment layers above extinct fluid conduits. We used the sediment thickness to infer an age at which deep fluid and gas flow induced by magmatic intrusions must have ceased. Our results contrast with findings by Lizarralde et al. (2010) who claim that thermogenic methane is still actively released in all places presented in their study. As detailed above, we do not disagree with Lizarralde et al. (2010) about the general mechanism. However, we disagree that all of the off-axis sites are presently active in the sense of hydrothermal systems (we discovered none) and that their lifetime has to be taken into account. We claim that this process only occurs during and for a certain time (depending on the lifetime of a sill-driven hydrothermal system) after the magmatic intrusions intruded in the sediment. How long this process really occurs still needs further investigation.

Specific comments:

We value the specific comments of the referee and would be glad to improve wording, definitions, and figures in case of a positive evaluation.

To the referee, the purpose of biomarker and heat flow analyses is unclear (L150 and L255 in referee comment).

The analyses of biomarkers and heat flow delivered fundamental knowledge about the origin of the carbonate and the heat distribution in the basin, respectively. Both analyses helped to reach the conclusion that high-temperature processes at the investigated sites are extinct.

In L491 the referee comments on the way of sediment-sill interaction.

We agree that after sill-emplacement, heat is the driving force to induce chemical reactions, as observed also in other regions (Cruse and Seewald, 2006; Ishibashi et al., 2014). If that conclusion is not understandable from our manuscript, we will clarify it.

In L599 the referee criticizes the section about timing in our manuscript.

Indeed, ages are only available for the carbonate sample and from the sedimentation rate. However, we approached the cessation of active thermogenic methane release by taking the sediment thickness above extinct conduits into account. Of course, the resulting time is only an approximation. As we stated in our manuscript and in the comments above, the lifetime of a magmatic system needs further investigation before conclusions of the timing of active methane release can be drawn.

References

Berndt, C., Hensen, C., Mortera-Gutierrez, C., Sarkar, S., Geilert, S., Schmidt, M., Liebetrau, V., Kipfer, R., Scholz, F., Doll, M., Muff, S., Karstens, J., Planke, S., Petersen, S., Böttner, C., Chi, W.-C., Moser, M., Behrendt, R., Fiskal, A., Lever, M. A., Su, C.-C., Deng, L., Brennwald, M. S. and Lizarralde, D.: Rifting under steam – how rift magmatism triggers methane venting from sedimentary basins, Geology, 44(9), 767–770, 2016.

Cruse, A. M. and Seewald, J. S.: Geochemistry of low-molecular weight hydrocarbons in hydrothermal fluids from Middle Valley, northern Juan de Fuca Ridge, Geochim. Cosmochim. Acta, 70(8), 2073–2092, doi:10.1016/j.gca.2006.01.015, 2006.

Ishibashi, J. I., Noguchi, T., Toki, T., Miyabe, S., Yamagami, S., Onishi, Y., Yamanaka, T., Yokoyama, Y., Omori, E., Takahashi, Y., Hatada, K., Nakaguchi, Y., Yoshizaki, M., Konno, U., Shibuya, T., Takai, K., Inagaki, F. and Kawagucci, S.: Diversity of fluid geochemistry affected by processes during fluid upwelling in active hydrothermal fields in the Izena Hole, the middle Okinawa Trough back-arc basin, Geochem. J., 48(4), 357–369, doi:10.2343/geochemj.2.0311, 2014.

Lizarralde, D., Soule, S. A., Seewald, J. S. and Proskurowski, G.: Carbon release by off-axis magmatism in a young sedimented spreading centre, Nat. Geosci., 4(1), 50–54, doi:10.1038/ngeo1006, 2010.

---

## Author Response (AR1)

Kiel, 06.07.2018

Dear Helge Niemann,

Thank you for considering our manuscript '*On the formation of hydrothermal vents and cold seeps in the Guaymas Basin, Gulf of California*' for publication in *Biogeosciences*.
We very much appreciate the insightful comments of the reviewers and have made substantial revisions in the manuscript accordingly. The most important changes include the revision of figures and tables, the rearrangement of section 4.1.1, discussing now first the hydrothermal anomalies, and the revision of section 4.3, in which we have clarified that we do not disagree with Lizarralde et al. (2010) in general, but only on the timing of thermogenic methane release. We shortened the manuscript where possible, clarified how our results differ from those of earlier studies, and emphasized the importance of the biological contribution to the geochemical and geophysical results.
We hope that you agree that our revision has substantially improved the manuscript and that you will find it fit for publication in *Biogeosciences*.

Yours Sincerely
Sonja Geilert

Response to Referee #1

General comments:

Reviewer comment: I would urge the authors to check the manuscript more carefully, there are numerous places with very obvious grammatical errors, unexplained abbreviations, and incorrect reference to the figures. Also, some of the long sentences and excessive use of comma make it difficult to read sometimes.

Reply: We proofread the manuscript and corrected errors, shortened sentences, and revised references. We hope that the manuscript is now well understandable and the content clearly stated.

Reviewer comment: The referee criticizes in his comment mostly our conclusion that hydrothermal activity has ceased in the Guaymas Basin deduced from the observation of dominantly seawater signatures in the pore fluids and biogenic methane emissions. The referee proposes that our data could also be interpreted in an opposite direction, namely that recent hydrothermal activity drives a shallow convection cell that draws seawater into the sediment.

Reply: At first, we do agree with Lizarralde et al. (2010) that hydrothermal activity in the Guaymas Basin was once driving seepage and an elevated thermogenic methane flux to the water column. We can also not exclude that there is still thermogenic methane released into the basin driven by off-axis sills. However, as all seep sites investigated in this study show predominantly seawater composition, a simple correlation of detected sills and active thermogenic methane release as done by Lizarralde et al. (2010) appears not to be feasible. Our data set shows that deep processes are extinct, at least at the investigated sites so that it is not unlikely that at other places deep processes are extinct as well. At least, it does not seem valid to assume that all other off-axis sills represent active hydrothermal systems. In order to calculate accurate thermogenic carbon fluxes, sill emplacement mechanisms like longevity and spatial distribution need further investigation. We emphasized these conclusions in lines 744-751 and 824-834 of the revised manuscript.

Reviewer comment: The referee claims that the heat provided by the Black Smoker field might '*decompose the organic matter and therefore explains the mostly biogenic methane source*'.

Reply: If organic matter is decomposed by an elevated heat source, than the isotopic signal would be indicative of this thermogenic source. The thermogenic $\delta^{13}C_{CH4}$ signal is relatively heavy (about -40 to -20‰) compared to the biogenic signal (< -55‰). All our (unaltered) $\delta^{13}C_{CH4}$ data falls in the biogenic field (see Fig. 8) and are thus not decomposed by thermogenic alteration. The lateral heat from the Smoker field might support and enhance biological processes but is not responsible for the isotopic signal as proposed by the referee.

Reviewer comment: The referee further criticizes that we just might not have detected the deep fluid phase as it is decoupled from the gas phase and might arrive later or at a different location. He proposes that seawater might have diluted the deep signal and that the young age of the authigenic carbonate would also support a recent seepage event.

Reply: We do not think that this is a likely alternative to the presented hypothesis. The known active hydrothermal systems from the southern (Von Damm et al., 1985) and northern (Berndt et al., 2016) rift axis in the Guaymas Basin emit hot fluids with clear evidence for high temperature fluid-rock interactions and thermogenic gas production. Such a fluid is not found at any of the seeps. Instead, we found pore water containing predominantly biogenic methane, but which is otherwise only slightly diagenetic altered from seawater. Biogenic methane formation is expected to occur within the uppermost tens to a few hundreds of meters below the seafloor at low temperatures. Methane-enriched pore water sourced in those depths should be likewise enriched in other products of organic matter degradation (e.g. $NH_4$), like found at the Slope Site (lines 532-540 and Fig. 4, 7). The fact that only biogenic methane is significantly enriched at the seep locations lets us conclude that methane gas is percolating through shallow sediments (even forming gas hydrates as observed at North Site) rising along pre-existing low permeability pathways formed by previous hydrothermal activity. The detection of elevated gas flows at the investigated seep sites confirms our visible observation of active seep sites at the seafloor, like microbial mats or clams (lines 596-601).

The young age of the carbonate supports our hypothesis of a decoupled gas and fluid phase as only ascending gas is needed to drive AOM and the formation of authigenic carbonates. Biomarker, $\delta^{13}C_{CH4}$, $\delta^{18}O_{CaCO3}$ and $^{87}Sr/^{86}Sr$ signatures clearly point to a formation in seawater at ambient temperatures. We agree with the referee of a recent seepage event, however, mainly driven by shallow-sources, biogenic gas and not by deep-sourced, hydrothermal processes. From sediment thicknesses above extinct fluid conduits we estimated that the processes must have stopped more than 7 kyrs ago at least at the places investigated so far. We cannot exclude that there are still areas in the Guaymas Basin with active sill-induced methane release. At our investigated sites though, we have not found any evidence of thermogenic methane release. A simple extrapolation as done by Lizarralde et al. (2010) in which they compile all sills and estimate the potential methane release appears not applicable.

We have emphasized that we indeed detected active seepage sites in lines 596-601 of the revised manuscript.

Reviewer comment: The referee claims that the title of our manuscript might be misleading as it appears to argue for the process of active thermogenic methane release. The referee also thinks that we should present our opinion already in the final sentences in the introduction (e.g., Line 97-99). At present, the referee thinks that we would agree with the transition from hydrothermal vents to cold seeps until later in the discussion.

Reply: The title of the manuscript might indeed be misleading. We referred to the processes itself and not to the activity. We changed the title to a more general meaning: '*On the formation of hydrothermal vents and cold seeps in the Guaymas Basin, Gulf of California*'. We also rephrased the last section of the introduction in order to clarify that we are comparing our findings to the hypothesis by Lizarralde et al. (2010) and added a sentence stating our findings (lines 116-123).

Specific comments:

Reviewer comment: Line 24: What does the 500m here mean?

Reply: 500 m is the distance to the Smoker field. Deleted in the abstract as it is more clearly defined in section 3.3 'Sediment characteristics and sedimentation rates'.

Reviewer comment: Line 31: If pore fluid is predominately seawater than you wouldn't call it "cold seep pore fluid".

Reply: 'Cold Seep' is a general term for areas where fluids, gases, and/ or solid material are transported from depth to the seafloor. Seepage often provides bioactive reductants like sulfide, methane, and hydrogen which fuel biota. This biota consists of typical cold seep communities like tubeworms, clams, and mussels and often occur with authigenic carbonates precipitated on the seafloor. The term 'cold' does not refer to the temperature of the seepage but is meant in contrast to 'hot' hydrothermal fluids.

In the Guaymas Basin, we have observed the typical seepage biota like mussels and clams as well as authigenic carbonate manifestations through video-guided MUC observations. Thus, we refer to this areas as 'cold seeps' as an active seepage area was identified.

We changed the geochemical definition of cold seep fluids from seawater to ambient diagenetic fluids in the abstract (lines 29-32 and 34-36), in lines 571-573, and in the Conclusions (line 841-844).

Reviewer comment: Line 48: Kennett 2000 is not a good citation as this paper only dealt with the Quaternary
excursions not PETM
Reply: We replaced Kennett (2000) with Aarnes et al. (2010), who discuss how contact metamorphism can trigger global climate change (line 60).

Reviewer comment: Line 74: delete "that"
Reply: Deleted (line 87)

Reviewer comment: Line 89: "a helium isotope signature indicative of mid-ocean ridge basalt." I guess you mean He isotopic signature tells them the fluid came from mid ocean ridge.
Reply: This sentence has been rephrased to indicate that fluids in contact with MORB were detected (lines 101-103).

Reviewer comment: Line 90: up to several hundred of meters.
Reply: The words 'up to' were added (line 104).

Reviewer comment: Line 91: "magmatic intrusions into underlying sediments" The orientation is weird in this sentence. Here the underlying should refer to the magmatic intrusion. Do you mean the intrusion penetrated strata deeper than it was?
Reply: We agree that the word underlying is confusing here and replaced it with deep (line 106).

Reviewer comment: Line: 94-97: Could you check the sentence again? If you intent to use two commas to form a clause, please remember to close the clause by adding the second comma. Also, consider using an active tone in this sentence, such as " during the SO241, we sampled at XXX and XXX locations."
Reply: The sentence was rephrased (lines 94-112).

Reviewer comment: Line 105: were
Reply: Changed to plural (line 130).

Reviewer comment: Line 106: check the articles of this sentence, not always "a"
Reply: Changed (lines 129-132).

Reviewer comment: Line 109: locations of seeps
Reply: Changed to plural (line 133)

Reviewer comment: Line 115: why need "respectively" here? What is GI gun?
Reply: The definition of the streamer was corrected from 'to' to 'and' as 'respectively' refers to 150m correlating with 96 channels and 183.5m correlating with 112 channels. GI gun was specified in the text (lines 138-140).

Reviewer comment: Line 126: I assume you mean authigenic carbonate concretions
Reply: The word 'carbonate' was added (line 150).

Reviewer comment: Line 127, 128: "Hence, comparing results from different seeps might be biased in this regard." Unclear what you mean.
Reply: We added the explanation that seepage areas might not been hit at the most active area (lines 152-153).

Reviewer comment: Line 131-134: The way you use comma is really confusing. For example, "at three seepage sites, North (GC01, MUC11),Central (GC03, GC13, GC15, MUC04), and Ring Seeps (MUC05)," Do you intent to say the three seepage sites include north, central and ring seeps sites? or the "three seepage site" is another site other than north, south, and Ring Seeps.
Reply: We have rephrased the sentence (lines 156-160).

Reviewer comment: Line 133: Are you sure you gave definition of the reference site "above" not "below"?
Reply: We corrected the definition to 'below' (line 158).

Reviewer comment: Line 155: "at a sampling rate of 1s." sampling rate of what?
Reply: The definition of sampling rate was defined as one measurement per second (line 189).

Reviewer comment: Line 171: "were"
Reply: Present is the correct tense here, so we did not change the word 'are' (line 206).

Reviewer comment: Line 187-193: I understand one can sure find details in the paper cited. However, I think it's important to mention things that are absolutely crucial. For example, it is important to mention how soon were the HS and ammonium analyzed after recovery of the porewater as both species are easily degraded due to oxidation and microbial consumption. It's also known that ammonium measurements by photometry method are heavily impacted by the presence of HS. What treatment did you do to prevent that. Titration of alkalinity is also a time-sensitive analyses as carbonate precipitation is still happening in the water samples. For the cation and anion samples brought back to shore lab, what preservation measure was performed. All of such information are crucial and I would like to see more description in the main text but not just "please refer to XXX".
Reply: We added a more detailed description of the methods. HS and $NH_4$ were analyzed right after core recovery and sampling. Before $NH_4$ analyses, the samples were treated with argon to expel HS. The pore fluids were acidified on board to inhibit mineral precipitation prior to shore-based elemental analyzes (lines 223-231).

Reviewer comment: Line 194-198: As volcanic material might be present in the study area, it is important to check the abundance of Rb and see if that affect the strontium isotopic ratios. This is supposed to be a routine for analyses like this. I would like to see some more information on this.
Reply: The potential impact of Rb interferences on Sr isotope ratios is avoided in multiple and independent steps as described below.
Based on prior Sr concentration measurements original sample aliquots typically equivalent to 1000 ng Sr were chemically separated for Sr after pre-treatment against potential organic content by single use highly selective Sr-Spec resin in a low blank one step chemistry. Usually, no significant amount of Rb is passing into the Sr eluate. However, a second physically purification is provided by measuring the isotope ratios on thermal ionisation mass spectrometry (TIMS). The lower ionisation temperature of Rb in comparison to Sr leads by slow heating and multiple focussing procedures on early Sr signals to preferential ionisation and depletion of potentially resin-passing traces of Rb.
The third and ultimate step to avoid any misleading data interpretation due to interfering 87Rb on 87Sr is the continuous monitoring of Rb abundances by measuring 85Rb in static mode simultaneously the Sr masses 84, 86, 87 and 88. This additional information was added in the manuscript in lines 243-247.

Reviewer comment: Line 209: VPDB needs to be explained
Reply: An explanation for VPDB was added (line 259).

Reviewer comment: Line 234: where in the supplement?
Reply: The text section in the supplement explaining the XRD measurements was indeed missing and was now added.

Reviewer comment: Line 231-241: This appears to be a ridiculously long sentence. Please revise the whole paragraph so that it's more readable.
Reply: We shortened and divided the sentence (lines 282-292).

Reviewer comment: Line 272: blankening? Blanking? and Line 272-273: Im not a geophysicist but I thought the blanking zone in seismic profile is due to gas/water (stuff with low density) instead of sediment mobilization?
Reply: Blanking is the correct term here. We added a more detailed explanation of the signal interpretation. Gas and/ or water can cause the signal blanking. In contrast, sediment mobilization can explain the observed deformed strata (lines 324-325).

Reviewer comment: Line 287-290: check the unit for 60, 15 mbsf. I think you mean ms. Also, explain what is mbsf.
Reply: The units were corrected and mbsf defined as meters below sea floor (lines 329-330).

Reviewer comment: Line 317: what do you "lower meter"? do you mean shallow in GCs?
Reply: Shell fragments occur in shallow depth in the GC. The sentence has been rephrased (lines 374-375).

Reviewer comment: Line 333: photometry method measures total hydrogen sulfide, S2-, HS-, and H2S.Please revise throughout the text. Why for some ions you specified their charge (like SO42-) for others you ignored the charge (NH4, Li, Mg)? Also, please revise alkalinity to total alkalinity (TA) for clarity throughout the text.
Reply: Alkalinity has been abbreviated with TA and all sulfide species with TH2S. The inconsistency in mentioning the charge was revised.

Reviewer comment: Figure 4: From the figure, the TA from GC07 could be as high as over 70 meg/L however the highest value listed in supplementary is only 65 meg/L. Could you check this again? Also, I suggest modify the scale of the plot. For example, it is really hard to see the changes in Mg and Li concentrations from the plot despite the 10% increase and decrease in concentrations of these two ions. The figure should be able to reflect these variations better.
Reply: Fig. 4 was changed by adjusting the colors and symbols of the plot in order to improve readability. The scales of Mg and Li were increased to visualize concentration changes. There was indeed a mistake with the TA scale, which was corrected. Highest TA concentrations of GC7 are 65 meg/L.

Reviewer comment: Line 347: revise to TA and total HS.
Reply: Revised to TA and TH$_2$S (see comment to line 333).

Reviewer comment: Line 350: I do not agree Mg and Li concentrations are similar to seawater for all sites, you apparent have higher Mg and Li in GC07
Reply: We added a detailed description of the concentration variations for GC01, GC07, and GC09 (lines 408-415).

Reviewer comment: Line 370: the lowest and highest values I can see are -26.5 and -88.2
Reply: The values were corrected (lines 435-436).

Reviewer comment: Line 373: I don't see any dD-CH4 value reported for Smoker unless you mean VCTD data, which is not from porewater
Reply: dD-CH4 values for the Smoker area are from VCTD sampling and stem from the hydrothermal plume. We have specified this in the text (lines 431-440).

Reviewer comment: Line 385: There is no VCTD09 in your data from supplementary and figure 6.
Reply: We clarified in the text that the temperature values for the water column above the hydrothermal field (VCTD09) are from Berndt et al. (2016) and added the data in the figure (lines 452-454 and 461-462).

Reviewer comment: Line 398: I wonder what kind of calcite it is, high-Mg or low-Mg calcite.
Reply: By the uncertainty related maximum deviation of Δd104 (< 0.01) the XRD spectrum identifies calcite with a Mg fraction below 3 % according to Goldsmith et al. (1961). We added this information in lines 467-469.

Reviewer comment: Line 403: isn't the reproducibility should be reported in the method section.

Reply: The reproducibility was also mentioned in the method section (lines 242-243) and is now deleted here in lines 473-474.

Reply: We added the He-isotope value in line 471. Indeed it was shown in Berndt et al. (2016) that hydrocarbons are composed of hydrocarbons produced by thermogenic organic matter degradation (degradation driven by the released heat of the magmatic intrusion) and abiogenic hydrocarbon formation (see Figure DR9 in Berndt et al. (2016), supplement). However, the largest amount of methane stems from thermogenic organic matter degradation. We slightly adjusted the text to clarify that this discussion is presented in detail in Berndt et al., 2016 (supplement) (line 473).

Reviewer comment: Line 425: Check the format of citation
Reply: The sentence has been rephrased (lines 493-496).

Reviewer comment: Line 434-435: If you look closely to the raw data, both Sr and Ca concentrations are 10% elevated compared to the seawater value. Also one of the only two 87Sr/86Sr values reported from GC09 shows significantly lower value from seawater values. This again emphasize the authors should really adjust the scale of the plot (Figure 4) to reflect these small but significant changes.
Reply: We adjusted the scale in Fig. 4 and 5 in order to visualize the concentration ranges as suggested by the reviewer. There are six $^{87}$Sr/$^{86}$Sr isotope signatures values for GC09 and two for GC10 (see Table S2). We forgot to plot GC10 in Fig. 5 which was now added.
We rearranged the discussion for section 4.1.1 in order to clarify where we have detected hydrothermal fluids and where seawater concentrations. Now, we are first discussing the hydrothermal signatures found in the deep core section of GC09 (>4m) and then the remaining sites which show seawater composition. Hydrothermal indicators are higher Li and lower Mg concentrations and $^{87}$Sr/$^{86}$Sr isotope signatures clearly point to a hydrothermal endmember ($^{87}$Sr/$^{86}$Sr = 0.7059) for pore fluids from GC09 (>4m).
The seawater composition of the remaining pore fluids (shallow pore fluids (<4m) from GC09, GC10 and MUCs) is interpreted as shallow convection cell drawing seawater into the sediment. We hope the rearrangement of this section clarifies our interpretation of the data (lines 512-575).

Reviewer comment: Line 445-446: again, if you look into the data clearly you would probably slightly change the conclusion here.
Reply: We do agree with the presence of hydrothermal fluids in some areas close to the hydrothermal smoker field and rearranged the discussion to clarify our statement (see also comment above). Indications of hydrothermal fluids are now discussed in lines 512-523.

Reviewer comment: Line 449-450: You only have one indicator, NH4, reported here. I don't think you can justify for all. Not to mention NH4 concentration is affected not only by organic matter degradation but also cation exchange.

Reply: In the newly arranged manuscript, other indicators of deep fluids like Mg or Li are now mentioned in the section before (lines 512-523). The $NH_4$ serves as an example for catagenetic or diagenetic breakdown of organic matter and helps to clarify that none of these process occur at the seep sites. We rewrote this section in order to clarify this process (lines 524-546). Cation exchange might be responsible for the elevated $NH_4$ concentrations, but is of minor importance in this region with a high organic precipitation rate.

Reviewer comment: Line 455. I don't see why is relevant to refer fig 3 here. I thought you mean fig4. Also, this sentence is so odd. I don't quite sure I get your point. How do you know it's high level of AOM but not just sulfate reduction+organic matter degradation, which is in line with your high TA and NH4 levels.

Reply: Indeed, Fig.3 was a wrong reference here and was changed to Fig.4 (line 539). We rearranged the sentence and the whole section in order to clarify that the Slope Site serves only as an example of how a deep diagenetic altered fluid might look like. As the seep site fluids do not show similar elevated concentrations of $NH_4$ as well as seawater-like Mg and Li concentrations we concluded that no deep fluid is reaching the surface here anymore (lines 524-551).

Sulfate reduction and organic matter degradation are processes of AOM which serves as an umbrella term here.

Reviewer comment: Line 459-460: Of course the data could be explained this way, but alternatively, if there is just no input of methane from the Smoker site (GC09, GC10), then one would expect exactly the same porewater profiles as reported here. The present data provide no justification of whether seawater convection exists or not at these coring sites.

Reply: We deleted the sentence in lines 459-460. The rearrangement of this section deals now with the question of convection in lines 546-551. The sentence is phrased as a hypothesis and provides an explanation for the observed pore fluid composition. As the sentence is formulated as an assumption and not as fact, we see no need to change it here. We also added additional references to clarify that such a convection cell is a phenomenon observed before at sedimented hydrothermal areas (line 549).

Reviewer comment: Line 461: now you mentioned the Li anomaly. I think this observation should be mentioned earlier in the text.

Reply: The discussion of the Li anomaly is moved to the beginning of the discussion in section 4.1.1 (see also comment above to lines 434-435).

Reviewer comment: Line 464-466: both Sr and Ca concentrations are also slightly elevated and the one 87Sr/86Sr value from GC09 is also significantly lower than seawater value.

Reply: There are six $^{87}Sr/^{86}Sr$ isotope signatures for GC09 and two for GC10 (Table S2) which are discussed in lines 515 to 517 (see also comment to lines 434-435).

Reviewer comment: Line 464: What is the cause of high Li? Hydrothermal solution (Line 465) or mineral composition (Line 462). If the authors think it's the latter, you should provide a explanation of the process and how.

Reply: We think that mixing with hydrothermal fluids are causing the Li anomaly and that the hydrothermal deposits found in the deep section of the core solely facilitate fluid circulation in contrast to the diatomaceous clay (now discussed in lines 513 to 521).

Reviewer comment: Line 473-474: I in general agree this conclusion but think this paragraph could be better integrated with the paragraph discussing the porewater data of Smoker site. Especially the statement here is in contradiction to the statement in Line 434-435.
Reply: We rearranged the discussion in section 4.1.1 as explained above in the comment to lines 434-435

Reviewer comment: Fig. 7: what is the x-axis of (A)? Also, how the mixing lines were determined in (a) and (b), especially in the log-log plot and log-linear plot. I think for the mixing lines should look differently the ones from the current plots.
Reply: Fig. 7 has been reduced to the upper plot ($NH_4$ vs Li) as the lower plot does not provide any new information. Further, the mixing line has been calculated following: $Li_{mix} = Li_{phase1} * f_1 + Li_{phase2} * f_2$, with $f_1 + f_2 = 1$. Phase 1 is the Li concentration of Guaymas Vent south (Von Damm, 1990) and phase 2 is the Li concentration of North Seep. The mixing proportions of $NH_4$ have been calculated accordingly. This formula has been added to the caption of Fig. 7. As the mixing line is not a linear regression, the look of it in the plot agrees with the used equation.

Reviewer comment: Line 491-494: The authors really need to work on this statement to get a self-consistent conclusion on this. See my earlier comments on this.
Reply: We revised the discussion as explained in the comment above to lines 434-435

Reviewer comment: Line 502 "(active?)" appears without context. Please clarify.
Reply: The blanking of the seismic profile indicates a fluid conduit, but the profile cannot differentiate between active fluid and/ or gas flow. As we only conclude later in this paragraph that the fluid and gas phases must have been decoupled we decided to put the 'active' in question. In order to eliminate misunderstandings, we deleted the question mark and put in question if it is fluid and/ or gas flow (lines 581).

Reviewer comment: Line 505-509: Since methane can also be generated through hydrothermal activity and even abiogenic processes, I don't see why organic matter degradation signal is necessarily expected.
Reply: The term 'thermogenic degradation of organic matter' also includes the process of degradation of organic matter and formation of hydrocarbons by additional heat provided by magmatic intrusions (see also comment to line 421). In the Gulf of California this process is indicated for deeply buried and shallow sediment where hydrocarbons are transported e.g. by hydrothermal circulation to the seafloor (e.g. Simoneit et al., 1988). Small gas contributions in hydrothermal fluids in the northern Guaymas Basin are derived from abiogenic methane formation which is indicated by $\delta^{13}C_{CH4}$ data and $^3He/^4He$ content (Berndt et al., 2016). This points to a deep magmatic (intrusion) source. However, hydrocarbon formation by abiogenic processes in hydrothermal circulation cells cannot be excluded here (e.g. McDermott et al., 2015; and discussions in Berndt et al., 2016, supplement). We added this information in lines 591-593.

**Reviewer comment:** Line 526: In my view, it's weird to see one calls Li as a major porewater constituent, as it's only less than 30 microM in the porewater.

**Reply:** Li is considered as a major indicator for high-temperature sediment-water interactions, as are Mg and Cl. We agree that it might be confusing in this context and rearranged the sentence (lines 599-603).

**Reviewer comment:** Line 531 as a tracer

**Reply:** Added

**Reviewer comment:** Line 565: what kind of oxidation of methane you are talking about? Aerobic or anaerobic?

**Reply:** The oxidation of methane is probably be affected by anaerobe microbial oxidation above the sulfate-methane transition zone utilizing sulfate and additional electron-acceptors like nitrate, manganese(IV) or iron(III) (e.g. Jørgensen, 2006). Edited in lines 658-661.

**Reviewer comment:** Line 566: AOM enriches DIC in 12C.

**Reply:** We replaced $CO_2$ with DIC in line 662.

**Reviewer comment:** Line 566-568: Im not sure how you this process you described can help explain your data. Besides, if you look into the Borowski et al (1997) paper, the paper is intent to explain why d13C-CH4 is actually counter-intuitively light in the AOM zone. It's true that AOM supposes to make the residual methane heavier in isotopic signature but this is not what usually observed and definitely not what Borowski et al intent to explain in their paper.

**Reply:** We changed the citing paper to Whiticar (1999), who is indeed describing the observed process more appropriate (line 664).

**Reviewer comment:** Line 570: for anaerobic methane oxidation. It's important to specify which oxidation.

**Reply:** The type of oxidation was specified (lines 658-661).

**Reviewer comment:** Line 601-629: In the argument against the conclusion by Lizarralde et al., how does the observation the authors had, a convection of seawater into the shallow sub-surface in the Smoker Sites, affect such argument. It is likely that seawater convection in the hydrothermal is a short-term and contemporary process, the geochemical signal happened to be capture by the current study. In this case, how do you actually use the observation of no geochemical signal to argue against the conclusion by Lizarralde et al. Besides, the convection of seawater in hydrothermal regions must be driven by seeping of fluid in the hydrothermal vents. If as the authors claimed, the porewater profiles are indicative to seawater convection, isn't that just confirmed the hydrothermal activity?

**Reply:** We do not deny the activity of the hydrothermal system in general. We just state that at the investigated seep sites no deep signal is detected and there are no indications of actively released thermogenic methane. We cannot exclude that this process occurs in other areas of the basin, however Lizarralde et al. (2010) calculated methane flux might be excessive (see also reply to general comment). We clarified this in lines 744-750 and 823-833.

Reviewer comment: Line 656: It's unexpected to see the authors show AOM reaction such late in the paper as they have talked about a lot earlier in the text. I suggest move part of this discussion when they use porewater profiles to infer intensive AOM activity.
Reply: We have moved the AOM reaction to section 4.1.2 (*Cold seeps*) in which this process is explained in detailed for the first time.

Reviewer comment: Line 677-680: I agree that the various lines of evidence from the carbonate suggest the recent formation but I don't see how do these support the conclusion "cessation of deep fluid and gas mobilization" the authors derived from porewater data. Isnt that the young ages from authigenic carbonate suggest a very recent seepage event? Since porewater profiles are probably contemporary signatures, can really conclude that the seepage has died just because they see nothing from the porewater profiles? Similar to my earlier comment, the "boring" and seawater-like porewater profiles were interpreted by the authors as due to seawater convection in the shallow subsurface. If this is true, how can the authors use this to say that the deep fluid migration has stopped?
Reply: The $\delta^{13}C_{CH4}$ data of the bulk carbonate overlaps with the $\delta^{13}C_{CH4}$ values in the associated pore fluids. No indicators of a deep signal have been found in the carbonate. Indeed, carbonate formation requires a recent seepage event; therefore we concluded beneath others that the fluid and gas phase must have been decoupled and only gases are rising to the surface and precipitate together with Ca as authigenic carbonate. From our data, we observe that no deep fluids, in contrast to gases, are rising to the surface at the investigated sites. We cannot exclude that this process might still occur at other seepage areas not investigated in this study. However, from our data, which show predominantly seawater concentrations at the investigated seepage sites, we can conclude that deep processes are extinct. Furthermore, no thermogenic methane was detected at the seepage sites. Summarizing, we can say, that the thermogenic carbon flux calculated by Lizarralde et al. (2010) might be overrated. We conclude that carbon flux extrapolations need to take the longevity of sill-introduced thermogenic carbon emissions into account. We revised this section of our discussion to clarify our conclusions (lines 799-803 and 823-833).

Reviewer comment: Line 696: what is s.a.
Reply: s.a. refers to 'see above', but is now deleted as the active CH4-emission period was just calculated in the section before

Supplement tables:

Reviewer comment: Please revise the units of mmol or micromol to mM and microM throughout the table.
There is no such unit.
The meaning of "-" in all the tables are unclear. Does it mean samples/analyses are not available or it is below detection limit. Especially for the table of d13C and dD of methane, not clear why sometimes there is not measurements of dD despite the high concentration. Also, it's not clear how "-" different from just a blank in the table.
Reply: The units were revised accordingly and the '−' and blanks in the table replaced with not determined (n.d.), below detection limit (b.d.l.), and not applicable (n.a.). The measurements for δD were carried out first to check for variations at each site. If there were variations in δD, more analyses have been conducted. In the case of GC07 where high methane concentrations are present, the δD values did not vary much (see Table S3). Therefore we decided not to perform any further measurements here.

Response to Referee #2

General comments:

Reviewer comment: Referee #2 has difficulties with three main aspects of our manuscript concerning the biological significance, the spatial coverage of sampling sites, and the new discoveries of our manuscript in contrast to earlier studies.
First, the referee claims that the biological aspects of our study are too small to get published in Biogeosciences.
Reply: The main findings and conclusions of our study are based on biological aspects, like the microbial signature of $\delta^{13}C$ data and the AOM-dominating biomarkers identified in the carbonate. The detected microbial signatures helped to identify that deep processes are extinct nowadays. The biological results support our geochemical and geophysical observations and form a key point of our discussion.
Additionally, we understand that the objective of this journal is to publish research which combines biological, chemical, and physical investigations and which highlights the interaction between them (*see homepage Biogeosciences*). Our manuscript combines all three aspects and emphasizes the importance of an interdisciplinary research approach to draw the best possible conclusions.
We emphasized the importance of the biological input to our study in the abstract (lines 36-42) and in the conclusions (lines 847-848). In general, the discussion of biological signals represents a considerable part of our manuscript, as shown in section 4.2 in lines 658-700, and section 4.3.2, lines 766-789.

Reviewer comment: Secondly, the referee expresses his concerns that the spatial coverage of our sampling sites is not sufficient to infer basin-wide phenomena.
Reply: Sample locations were chosen based on findings by Lizarralde et al. (2010) who describes sill intrusions associated with hydrocarbon gas emissions, biological communities, and authigenic carbonates. In this study, we investigated 3 seepage sites at various distances from the hydrothermal vent field based on locations identified by Lizarralde et al. (2010) as areas of active methane release. Additionally, a reference site, smoker sites as well as the water column have been sampled. With the exception of the active smoker site, there is no indication for a deep fluid advection and methane $\delta^{13}C$ data are predominantly of microbial origin (see Fig. 4 and section 4.4). Despite the fact that no deep fluids were detected at the seepage sites, an active methane flux was present, indicating that we hit the currently active sites described in Lizarralde et al. (2010). The detected methane was predominantly of microbial origin and no active thermogenic methane is released nowadays at the investigated sites as claimed by Lizarralde et al. (2010). We cannot exclude the possibility that thermogenic methane is still released in other areas of the basin, but the lack of evidence for high temperature geochemical processes at the investigated sites contradicts with Lizarralde's et al. (2010) conclusions. The seismic evidence of seep-induced hydrothermal systems alone is not sufficient for projecting methane emissions for the whole basin at present (see also comment to referee#1). Thermogenic methane release induced by off-axis sill intrusions is still a likely process to occur, but our study suggests that the lifetime of these systems is limited and has to be taken into account for budget calculations. Hence, the study of Lizarralde remains valid and is highly valuable in terms of describing the general process and the potential magnitude, but care has to be taken concerning the longevity of the hydrothermal systems and associated thermogenic methane release after the occurrence of sill intrusions.

We clarified this section of our discussion and explained the applicability of our results to the whole basin (lines 744-750 and 823-833).

Reviewer comment: The last major point of criticism by the referee is that it is not clear how the findings of this study differ from those of Lizarralde et al. (2010) and Berndt et al. (2016).

Reply: While Berndt et al. (2016) focused on characterizing the geophysical and geochemical characteristics of the Smoker area, Lizarralde et al. (2010) investigated geophysical aspects of the wider basin and the water column. Our study is the first one to look at geochemical, biological, and geophysical characteristics of seepage sites and the water column above. Main findings are the decoupling of gas and fluid phases, the microbial origin of methane, and the detection of sediment layers above extinct fluid conduits. We used the sediment thickness to infer an age at which deep fluid and gas flow induced by magmatic intrusions must have ceased. Our results contrast with findings by Lizarralde et al. (2010) who claim that thermogenic methane is still actively released in all places presented in their study. As detailed above, we do not disagree with Lizarralde et al. (2010) about the general mechanism. However, we disagree that all of the off-axis sites are presently active in the sense of hydrothermal systems (we discovered none) and that their lifetime has to be taken into account. We claim that this process only occurs during and for a certain time (depending on the lifetime of a sill-driven hydrothermal system) after the magmatic intrusions intruded in the sediment. How long this process really occurs still needs further investigation.

We emphasized our study results in contrast to Berndt et al. (2016) and Lizarralde et al. (2010) throughout the whole manuscript, e.g. in the introduction (lines 109-123), in section 3.6 (lines 461-462), in section 4.1 (lines 488-495), in section 4.2 (lines 647-648 and 678-681), and in lines 744-750.

Specific comments:

Reviewer comment: L001: I think more specific wording describing what authors observed seems better.

Reply: We have changed the title according to the suggestion by Referee #1.

Reviewer comment: L021: This sentence seems inadequate as abstract of this study.

Reply: We have rearranged the sentence to a more introductory meaning and emphasize the motivation of our study (lines 23-26).

Reviewer comment: L024: In a research field for hydrothermal activity, horizontal distance of _500m is not "close". See Cruse&Seewald 2006; 2010; Reeves et al. 2011; Baumberger et al. 2016; or some other numerous papers.

Reply: In 'close distance' is meant here relatively to the other investigated sites. Unfortunately, it was not possible to obtain samples closer to the hydrothermal vent field as sediment composition did not allow core penetration. It is true that compared to other hydrothermal areas 500m is not close.  We added relatively here to emphasize this (line 26).

Reviewer comment: L040: Introduction, carbon flux from seafloor to atmosphere, is not closely related to what authors observed in THIS study.

Reply: Indeed, our observations do not show a (thermogenic) carbon flux from the seafloor to the atmosphere. However, the aim of our study was to investigate the causes of global warming, e.g. during the PETM. One hypothesis is that magmatic intrusions into organic-rich sediments might release large amounts of thermogenic methane which might have triggered climate warming. Based on this theory, Lizarralde et al. (2010) studied water column anomalies above potential seepage areas in the Guaymas Basin and concluded that large amounts of thermogenic methane are still released today.  Lizarralde et al. (2010) inferred therefore that magmatic intrusions might have triggered the climate warming during the PETM. In contrast, our detailed study of pore fluids and gases of the seepage areas mentioned in Lizarralde et al. (2010) did not show active thermogenic methane release or rising of deep fluids. We concluded therefore that the methane release calculated by Lizarralde et al. (2010) might be too high.
Our study investigates processes possibly responsible for climate warming and therefore we think that we can begin our introduction with introducing this hypothesis.

Reviewer comment: L040: L051: This paragraph can move to M&M.

Reply: We do not agree that this paragraph should move to the *Materials & Method* section as it provides background information on the geological setting of the sampling area. The geological characteristics of the Guaymas Basin and the composition of the sediments are explained. As these are no new information gathered in this study, we concluded to describe them in the introductory paragraph. We will not move this section to the *Materials & Method* section as this section should only give information about samples investigated in this study and methods applied here.

Reviewer comment: L069: What is environmental conditions?

Reply: Environmental conditions refer to the enhancement of early-diagenetic reactions and with that the distinct changes in fluid and gas geochemistry. We specified environmental conditions with early-diagenetic processes in the main text (lines 80-83).

Reviewer comment: L071: Magmatic intrusion is geological process while fluid-rock and fluid-sediment interactions (associated with magmatic heat) influences fluid/sediment geochemistry. Because major part of this study is geochemical description, it seems better to make the wordings clear.

Reply: We agree that the sentence is imprecise and we defined now that the heat released by the magmatic intrusions is causing the fluid chemistry to change by accelerating early-diagenetic processes (lines 83-86).

Reviewer comment: L075: These sentences (L075-082) seem inadequate for this study.

Reply: We do not agree that these sentences are inadequate for this study as the process described by Lizarralde et al. (2010) was our motivation to conduct this study. Our reason to investigate these seepage sites was to study pore fluid and gas signatures influenced by magmatic-induced early-diagenetic reactions. Even though our results revealed that deep processes are extinct at the investigated sites, we think it is appropriate to introduce

Lizarralde et al. (2010) theory. Therefore we will leave the overview of Lizarralde et al.'s (2010) finding at the end of the introduction.

Reply: Seismic data was acquired in this study and we clarified this in lines 115-116, 136-143, 319-344.

Reply: We have changed the definition of the mat to microbial (line 149).

Reply: The names of the samples refer to the type of core we retrieved as GC for gravity corer and MUC for multi-corer. We think that the names are appropriate as they indicate for the reader the core type and depth of the sample at once.  We prefer to leave the naming of the samples as they are.

Reply: We replaced sampled with subsampled (line 164).

Reply: A reference was added in line 165.

Reply: The difference of MUC core retrieval in contrast to GC core retrieval is described in the following lines (former manuscript lines 141-144). In contrast to GC samples, MUC samples were brought to a cooling lab and sampling was executed anoxic in an argon-flushed glove bag. Retrieved pore fluids were centrifugation and subsequent filtered. Explained in the revised manuscript in lines 167-171.

Reply: A reference was added in line 171.

Reply: The heat flow measurements delivered fundamental knowledge about the heat distribution in the basin and helped to characterize the influence of the hydrothermal vent field and the sill-intrusions. As the heat flow significantly drops further away from the hydrothermal vent field, the heat flow analyses helped to support our hypothesis that deep processes are extinct. The intruded sills are no longer releasing heat which might accelerate early-diagenetic processes.

Reply: The names of the samples from the water column are following the same principle as the pore fluid samples. The name indicates the station name and in brackets we indicate the type of instrument used. We think that this way of naming is reasonable and we see no need to change it.

Reply: Indeed, the 2 was missing in the name of the instrument (Thermo MAT 253) and it was added in line 257.

Reply: The sample was freeze dried sediment. We added this information in the text in lines 267-268.

Reply: The analyses of biomarkers was providing (similar to the heat flow measurements, see above) fundamental knowledge about the origin and characteristics of the carbonate. The biomarkers showed a clear AOM origin, which supported our hypothesis that deep processes are extinct and that the methane needed to form the carbonate stems from shallow AOM processes.

Reply: 'Raw' vertical profiles are now shown in the supplement, Fig. S2.

Reply: The water column chemistry was investigated and reported first in this study except for the one water column directly above the hydrothermal vent field (VCTD09), which was reported first in Berndt et al. (2016). We emphasized this in the result section now in lines 422-426.

Reply: We indeed confused the section numbering here and corrected it for 4.1

Reply: The water column data directly above the hydrothermal vent field stems from Berndt et al. (2016). We clarified this in the text in lines 452-456 and 460-461.

Reply: The sentence is formulated as an assumption and we see no reason to change it, as it simply provides a possible explanation for the observed high heat flow.

Reply: Additional studies which observed convection cells in sedimentary basins are Gamo et al. (1991) and Kinoshiita and Yamano ( 1997). We added these references in the text in line 548.

Reply: Fig. 3 is a wrong reference here and we changed it to Fig. 4 (line 538).

Reviewer comment: L491: I guess chemical reactions between sediment and intruded sill occur only at the time of eruption event. Fluid-sediment interaction associated with magmatic heat source occurs more likely. See Cruse&Seewald 2006 GCA, Ishibashi et al. 2014 Geochem.J, or some other papers reporting fluid geochemistry of sediment-covered vent sites.
Reply: We agree that after sill-emplacement, heat is the driving force to induce chemical reactions, as observed also in other regions (Cruse and Seewald, 2006; Ishibashi et al., 2014). We clarified this in the text in lines 567-569.

Reviewer comment: L599: The story of timing of methane release seems frail due to limited evidences for temporal scaling. Information about time is only derived from solid phase (carbonate geochronology and sedimentation rate), and no evidence about past methane release is presented. Although past intrusion into sediment suggested by seismic dataset may imply generation and and release of thermogenic methane at the time of intrusion, it is just interpretation.
Reply: Age data is in fact only available for the carbonate sample and can be deduced from the sedimentation rate. However, we approached the cessation of active thermogenic methane release by taking the sediment thickness above extinct conduits into account. Of course, the resulting time is only an approximation. As we stated in our manuscript and in the comments above, the lifetime of a magmatic system needs further investigation before conclusions of the timing of active methane release can be drawn (lines 743-749 and 822-832).

Reviewer comment: L703: This is not conclusion of this study.
Reply: We provided this information as an explanation for the motivation of our study. We think it is justified to provide this information here and see no need to change it.

Reviewer comment: L712: This interpretation has been clear before this study and is not proved in this Study
Reply: Seismic data acquired in this study clearly showed that fluid and gas conduits above sill-intrusions were active once. From pore fluid geochemical data we can deduce that no deep processes are acting anymore. We have proven in our study that the longevity of the magmatic system is a crucial factor which needs to be taken into consideration when interpolating active methane release. From sediment thicknesses above extinct conduits we deduced the time, when hydrothermal circulation must have stopped at the seep sites (lines 806-820). Therefore we think that this sentence is justified at the end of the conclusions.

Reviewer comment: Fig1: Not informative. Except DSDP site and zoom up for seep-vent sites are better.
Reply: We plotted the DSDP site in Fig. 1 as it is our geochemical reference site for the hydrothermal endmember described by Von Damm et al. (1985) and Von Damm (1990). Therefore we prefer to leave the DSDP site in our map. In order to improve the visibility of the seep and smoker stations, we added enlargements here.

Reviewer comment: Fig3: Y-axis scaling is not good. Using two panels for large and small heat flows is better.

Reply: We changed the appearance of Fig.3 and hope that the visibility of the heat flow distribution has now improved. We added an extra panel and scale for the high heat flow for the rift valley and Smoker Site.

[revised manuscript text omitted]

---

## Editor Decision (ED1)

[revised manuscript text omitted]

fig02

[Figure]

fig03

[Figure]

fig04

[Figure]

fig05

[Figure]

fig06

[Figure]

fig07

[Figure]

fig08

[Figure]

fig09

[Figure]

---

## Author Response (AR2)

Kiel, 04.09.2018

Dear Helge Niemann, please find below our changes to the manuscript '*On the formation of hydrothermal vents and cold seeps in the Guaymas Basin, Gulf of California*' for publication in *Biogeosciences*.
We have substantially edited the abstract, the introduction, and the conclusions to improve the understanding of our main message. Further, the manuscript has been edited for its tenses, changing the method and result sections to past tense. We have improved the appearance and uniformity of the figures.

**Specific comments**

Line 136: a reference was added in line 191 of the marked up manuscript
Line 138: the sentence was rephrased in lines 191-194
Line 139: the sentence was rephrased in line 194-196
Line 140: the sentence was rephrased in line 196-198
Line 149: changed to cold lab in line 208
Line 151: a detailed description of the hydrocarbon sampling procedure was added in lines 215-223
Line 158: all figure captions were moved to the end of the manuscript
Line 184: the sentence was rephrased in lines 251-254
Line 199: the sentence was rephrased in lines 266-269
Line 207: the term was rephrased in line 276
Lines 240-242: the sentence was rephrased in lines 309-311
Lines 248-249: Information on the reference materials was added in lines 318-322
Line 270: the word 'analyses' was added in line 343
Line 284: a reference was added in line 361
Line 366: see comment to line 158
Lines 585-588: The sentence was rephrased and the reference deleted in lines 647-650

In general the manuscript has been edited for readability and comprehensibility.

We hope that you agree that our revision has substantially improved the manuscript and that you will find it fit for publication in *Biogeosciences*.

Yours Sincerely
Sonja Geilert

**On the formation of hydrothermal vents and cold seeps in the Guaymas Basin, Gulf of California**

Sonja Geilert[1], Christian Hensen[1], Mark Schmidt[1], Volker Liebetrau[1], Florian Scholz[1], Mechthild Doll[2], Longhui Deng[3], Annika Fiskal[3], Mark A. Lever[3], Chih-Chieh Su[4], Stefan Schlömer[5], Sudipta Sarkar[6], Volker Thiel[7], Christian Berndt[1]

[1]GEOMAR Helmholtz Centre for Ocean Research Kiel, Wischhofstraße 1-3, 24148 Kiel, Germany
[2]Universität Bremen, Klagenfurter-Straße 4, 28359 Bremen, Germany
[3]Department of Environmental Systems Science, ETH Zurich, Universitätstrasse 16, 8092 Zurich, Switzerland
[4]Institute of Oceanography, National Taiwan University, No. 1, Sec. 4, Roosevelt Road, Taipei 106, Taiwan
[5]Federal Institute for Geosciences and Natural Resources, Stilleweg 2, 30655 Hannover, Germany
[6]Department of Earth and Climate Science, Indian Institute of Science Education and Research Pune, Dr. Homi Bhabha Road, Maharashtra-411008, India
[7]Geobiology, Geoscience Centre, Georg-August University Göttingen, Goldschmidtstr. 3, 37077 Göttingen, Germany

**Abstract**

Magmatic sill intrusions into organic-rich sediments cause the release of thermogenic $CH_4$ and $CO_2$. Pore fluids from the Guaymas Basin (Gulf of California) - a sedimentary basin with recent magmatic activity – were investigated to constrain the link between sill intrusions and fluid seepage as well as the timing of sill-induced hydrothermal activity. Sampling sites were close to a hydrothermal vent field at the northern rift axis and at cold seeps located up to 30 km away from the rift. Pore fluids close to the active hydrothermal vent field showed a slight imprint by hydrothermal fluids and indicated a shallow circulation system transporting seawater to the hydrothermal catchment area. Geochemical data of pore fluids at cold seeps showed a mainly ambient diagenetic fluid composition without any imprint related to high temperature processes at greater depth. Seep communities at the seafloor were mainly sustained by microbial methane, which rose along pathways formed earlier by hydrothermal activity, driving anaerobic oxidation of methane (AOM) and the formation of authigenic carbonates.

Overall, our data from cold seep sites suggest that sill-induced hydrothermalism is not active away from the ridge axis at present and vigorous venting of hydrothermal fluids is restricted to the ridge axis. Using the sediment thickness above extinct conduits and carbonate dating, we calculated that deep fluid and thermogenic gas flow ceased 28 to 7 kyrs ago. These findings imply a short lifetime of hydrothermal systems limiting the time of unhindered carbon release as suggested in previous modeling studies. Consequently, activation and deactivation mechanisms of these systems need to be better constrained for the use in climate modeling approaches.

The Guaymas Basin in the Gulf of California is an ideal location to investigate the hypothesis that magmatic intrusions into organic-rich sediments can cause the release of thermogenic methane and $CO_2$ which may contribute to climate warming. In this study pore fluids relatively close to a hydrothermal vent field and at cold seeps up to 30 km away from the northern rift axis were studied to determine the influence of magmatic intrusions on pore fluid composition and gas migration. Pore fluids close to the hydrothermal vent field show predominantly ambient diagenetic fluid composition, indicating a shallow circulation system transporting seawater to the hydrothermal catchment area rather than being influenced by hydrothermal fluids themselves. Only in the deeper part of the sediment core, composed of hydrothermal vent debris, $^{87}Sr/^{86}Sr$ ratios and slightly elevated Li concentrations indicate the minor admixture of hydrothermal fluids (~3%). Pore fluids at cold seeps also show a mainly ambient diagenetic fluid composition without any imprint from high temperature processes. Seep communities at the seafloor are mainly sustained by biogenic methane, which is rising along pre-formed pathways. Anaerobic oxidation of methane (AOM) is widespread at these sites as indicated by pore water profiles, isotope fractionation of hydrocarbons, as well as the occurrence of authigenic carbonates and indicative biomarkers.

Deep fluid and thermogenic gas flow might have been active during sill emplacement at the investigated sites, but ceased 28 to 7 kyears ago, based on sediment thickness above extinct conduits. Our results indicate that carbon release depends on the longevity of sill-induced hydrothermal systems, which is a currently unconstrained factor.

**1 Introduction**

Climate Abrupt climate change events in Earth's history have been partly related to the injection of large amounts of greenhouse gases into the atmosphere (e.g. Svensen et al., 2004; Gutjahr et al., 2017). Among the most prominent of these events was the Paleocene-Eocene Thermal Maximum (PETM) during which the Earth's atmosphere warmed by about

8°C in less than 10,000 years (Zachos et al., 2003). The PETM was possibly triggered by the emission of about 2000 Gt of carbon (Dickens, 2003; Zachos et al., 2003). Processes discussed to release these large amounts of carbon in a relatively short time are gas hydrate dissociation, volcanic eruptions as well asand igneous intrusions into organic-rich sediments, triggering the release of carbon during contact metamorphism (Svensen et al., 2004; Aarnes et al., 2010; Gutjahr et al., 2017; Svensen et al., 2004)).

The Guaymas Basin in the Gulf of California is considered as one of the few key sites to study carbon release in a rift basin exposed to high sedimentation rates. A newly discovered vent field in the Guaymas Basin, which releases large amounts of $CH_4$ and $CO_2$ up to several hundred of meters into the water column (Berndt et al., 2016), stimulated the discussion on the climate potential of magmatic intrusions into organic-rich sediments (e.g. Svensen et al. 2004).

The Gulf of California is located between the Mexican mainland and the Baja California Peninsula, north of the East Pacific Rise (EPR; Fig. 1). The spreading regime at EPR continues into the Gulf of California and changes from a mature, open ocean-type to an early-opening continental rifting environment with spreading rates of about 6 cm $yr^{-1}$ (Curray & Moore, 1982). Its spreading axis consists of two graben systems (northern and southern troughs) offset by a transform fault (Fig. 1). The Guaymas Basin, which is about 240 km long, 60 km wide, and reaching water depths of up to 2000 m, is known as a region of vigorous hydrothermal activity (e.g. Curray and Moore, 1982; Gieskes et al., 1982; Von Damm et al., 1985). Hydrothermal activity in the Guaymas Basin was first reported in the southern trough (e.g. Lupton, 1979; Gieskes et al., 1982; Campbell and Gieskes, 1984; Von Damm et al., 1985). Here, fluids emanate partly from Black Smoker type vents at temperatures of up to 315°C (Von Damm et al., 1985). The rifting environment in the Guaymas Basin shows a high sediment accumulation rate of up to 0.8-2.5 m $kyr^{-1}$ resulting in organic-rich sedimentary deposits of several hundreds of meters in thickness (e.g. Calvert, 1966; DeMaster, 1981; Berndt et al., 2016). The high sedimentation rate is caused by high biological productivity in the water column and influx of terrigenous matter from the Mexican mainland (Calvert, 1966). Sills and dikes intruding into the sediment cover significantly affecthave a substantial impact on the distribution of heat flow, temperature distribution and henceother environmental conditions and thuslike early-diagenetic processes within the basin (Biddle et al., 2012; Einsele et al., 1980; Kastner, 1982; Kastner and Siever, 1983; Simoneit et al., 1992;

Lizarralde et al., 2010; Teske et al., 2014).

Magmatic intrusions and cold seeps at the seafloor were observed up to 50 km away from the rift axis, and a recently active magmatic process triggering the alteration of organic-rich sediments and releasing thermogenic $CH_4$ and $CO_2$ was proposed   by Lizarralde et al, (2010). These authors attributed elevated $CH_4$ concentrations and temperature anomalies in the water column result from active thermogenic $CH_4$  production driven by contact metamorphism. According to Lizarralde et al. (2010) ongoing off-axis hydrothermal activity may cause a maximum carbon flux of 240 kt C $yr^{-1}$ through the seafloor into the ocean and  potentially into the atmosphere. However, modelling studies investigating the lifetime of such sill-induced hydrothermalism show that initial $CH_4$ and $CO_2$ release is intense and vigorous, but can decline just as quickly (<10 kyr) (Bani-Hassan, 2012; Iyer et al., 2017).

[revised manuscript text omitted]

**2.1.2 Sediment and pore fluid sampling**

At seepage and vent sites, the video-guided MUC was used to discover recent fluid release, which  was indicated by typical chemosynthetic biological communities at the seafloor (microbial mats, bivalves, etc.; Sahling et al., 2002). However, small-scale, patchy distributions of active seepage spots and visibility of authigenic carbonate concretions made it difficult to select the best locations for coring. Hence, the comparing of results from different seeps might be biased in this regard as not all seepage areas could be sampled at their most active places. GC

deployments were typically performed at sites initially investigated with the MUC video system or at the center of suspected seeps (based on bathymetry and seismic data).

In total, we present pore fluid and gas data collected at the seepage sites North (GC01, MUC11), Central (GC03, GC13, GC15, MUC04), and Ring Seep (MUC05), one reference site (see below; Reference Site; GC04, MUC02), and the hydrothermal vent field (Smoker Site; GC09, GC10, MUC15, MUC16). The Reference Site, that did not show active seepage or faults indicated by seismic data, was chosen to obtain geochemical background values. In addition, the slope towards the Mexican mainland was sampled as well (Slope Site; GC07) (Fig. 1, Table 1). Immediately after core retrieval, GCs were cut, split, and subsampled. Samples were transferred into a  cold lab at 4°C and processed within 1 or 2 hours. Pore fluids were obtained by pressure filtration (e.g. Jahnke et al., 1982).  After MUC retrieval, bottom water was sampled and immediately filtered for further analyses. The sediment was transferred into a  cold lab and sampling was executed in an argon-flushed glove bag. Pore fluids were retrieved by centrifugation and subsequent filtration using 0.2 µm cellulose acetate membrane filters (e.g. Jahnke et al., 1982). Sediment samples (2 cm$^3$) for hydrocarbon analyses were taken using cut-off 3-mL syringes. All hydrocarbon samples were taken immediately after sediment surfaces were exposed after core cutting or sectioning, ensuring minimal disturbance to sediment surfaces prior to sampling and transferred to vials containing concentrated NaCl solution (after Sommer et al., 2009). MUCs were extruded and sampled from the top. GCs were sampled at the bottom ends of 1-m core sections, either at the core catcher or at freshly cut section ends. In some cases additional samples were taken from within GC core sections by cutting the core liner with an oscillating saw, and inserting cut-off syringes into the sides of core sections.

[revised manuscript text omitted]
. 2016) and the widespread occurrence of sills and fluid escape features within the basin has been used to estimate the related carbon release (Lizarralde et al. 2010). Our investigations of off-axis methane seeps in the Guaymas Basin demonstrate that there are no indications for hydrothermal activity away from ridge axis at present. These conclusions are mainly based on the lack of geochemical signals from high temperature alteration processes and $CH_4$ predominantly originating from microbial degradation. We suggest that hydrothermal circulation has, based on seismic records and dating of authigenic carbonates, largely ceased at the investigated sites roughly some thousand years ago. This finding underlines that the vigorous venting, as presently observed at the ridge axis, is a very effective way to release sedimentary carbon into the water column, but must be considered as a very short-lived process in a geological sense. Hence, a more comprehensive understanding of these hydrothermal systems with respect to their climate relevance requires a better knowledge on the control mechanisms and their longevity.

[revised manuscript text omitted]